# Neural stem cells induce the formation of their physical niche during organogenesis

Ali Seleit[1,2†], Isabel Krämer[1,2†], Bea F Riebesehl[1], Elizabeth M Ambrosio[1], Julian S Stolper[1,3], Colin Q Lischik[1,2], Nicolas Dross[4], Lazaro Centanin[1*]

[1]Animal Physiology and Development, Centre for Organismal Studies Heidelberg, Heidelberg, Germany; [2]The Hartmut Hoffmann-Berling International Graduate School of Molecular and Cellular Biology, University of Heidelberg, Heidelberg, Germany; [3]Murdoch Childrens Research Institute, University of Melbourne, Melbourne, Australia; [4]Nikon Imaging Center at the University of Heidelberg, Heidelberg, Germany

*For correspondence:
lazaro.centanin@cos.uni-heidelberg.de

†These authors contributed equally to this work

Competing interests: The authors declare that no competing interests exist.

**Abstract** Most organs rely on stem cells to maintain homeostasis during post-embryonic life. Typically, stem cells of independent lineages work coordinately within mature organs to ensure proper ratios of cell types. Little is known, however, on how these different stem cells locate to forming organs during development. Here we show that neuromasts of the posterior lateral line in medaka are composed of two independent life-long lineages with different embryonic origins. Clonal analysis and 4D imaging revealed a hierarchical organisation with instructing and responding roles: an inner, neural lineage induces the formation of an outer, border cell lineage (nBC) from the skin epithelium. Our results demonstrate that the neural lineage is necessary and sufficient to generate nBCs highlighting self-organisation principles at the level of the entire embryo. We hypothesise that induction of surrounding tissues plays a major role during the establishment of vertebrate stem cell niches.

DOI: https://doi.org/10.7554/eLife.29173.001

## Introduction

Animal organs are composed of diverse cell types, which in most cases derive from different embryonic origins. The last decades have witnessed huge efforts focused on identifying molecules necessary for organ formation in a wide range of organisms. This approach resulted in a deeper understanding of the molecular networks that control organogenesis in both vertebrates and invertebrates (*Gilbert, 2014*). Even though a broad range of molecular details have been uncovered, those did not contribute to revealing how cells from different embryonic origins end up together in the same functional unit.

There are two alternative scenarios on how cells from different lineages successfully come together during embryogenesis to assemble composite organs. One option is that they originate from cells that form in remote areas and migrate to a common pre-defined location. This typically happens during the formation of gonads in vertebrates, where germ cells follow a directional migration towards the somatic cells that will form their stem cell niche and eventually the gonad (*Doitsidou et al., 2002*; *Santos and Lehmann, 2004*). The other option is that one cell type is generated in a stereotypic position to then recruit a second cell type to fulfil a new role *in situ*. This occurs during eye formation, where retinal progenitors trigger the induction of surface ectodermal cells into lens cells that are later integrated into the forming eye (*Cvekl and Ashery-Padan, 2014*; *Gunhaga, 2011*).

While gonads, eyes and most animal organs have a defined number and location within species, the neuromasts - sensory organs of the lateral line system in fish - constitute a more plastic model in

which organ numbers cannot exactly be deduced from the developmental stage or overall size (*Ghysen and Dambly-Chaudière, 2007*; *Seleit et al., 2017*). The first neuromasts of the posterior lateral line (pLL) system are generated during embryogenesis by one primordium (in medaka) and two primordia (in zebrafish, tuna and *Astyanax*) that migrate anterior-to-posterior along the horizontal myoseptum while depositing groups of cells from their trailing ends at regular intervals. Soon after deposition these cellular clusters mature into neuromasts, composed of three main cell types as assessed by morphology and molecular markers. Hair cells (HCs), the sensory component and differentiated cell type, project cilia into the external environment and are responsible for detecting water flow and relaying this information back to the CNS (*Ghysen and Dambly-Chaudière, 2007*; *Williams and Holder, 2000*). Surrounding the HCs is a population of progenitor support cells (SCs) (*Ghysen and Dambly-Chaudière, 2007*; *Hernández et al., 2007*), while mantle cells (MCs) form an outer ring around both cell types and encapsulate the projecting HCs in a cupula-like structure (*Jones and Corwin, 1993*; *Steiner et al., 2014*). Since HCs are continuously lost and replaced under homeostatic conditions (*Cruz et al., 2015*; *Williams and Holder, 2000*) and in response to injury (*Hernández et al., 2007*; *López-Schier and Hudspeth, 2006*) throughout the life of fish (*Pinto-Teixeira et al., 2015*), the existence of stem cells capable of constantly replenishing HCs has been proposed.

The presence of neuromast stem cells has also been inferred from the fact that new neuromasts are continuously added to the lateral line system during the lifetime of fish, presumably to cope with an increasing body size (*Dufourcq et al., 2006*; *Ghysen and Dambly-Chaudière, 2007*; *Wada et al., 2013*). It has been reported that the peripheral ring of mantle cells plays a major role in new organ formation (*Dufourcq et al., 2006*; *Jones and Corwin, 1993*; *Stone, 1933*; *Wada et al., 2013*). This has led to the proposition of a linear model in which mantle cells give rise to support cells that in turn replenish lost hair cells (*Ghysen and Dambly-Chaudière, 2007*). The transition from support to hair cells has been heavily investigated in the past decade and is strongly supported by a large body of experimental evidence (*Hernández et al., 2007*; *López-Schier and Hudspeth, 2006*; *Ma et al., 2008*; *Romero-Carvajal et al., 2015*; *Wibowo et al., 2011*; *Williams and Holder, 2000*). However, the proposed transition of mantle into support cells has not yet been shown and it is still unknown where the stem cells of the neuromasts reside (*Pinto-Teixeira et al., 2015*). An interesting observation is that upon repeated injury of the hair cell population coupled with a depletion of support cells, mantle cells re-enter the cell cycle (*Romero-Carvajal et al., 2015*), but their precise role during homeostasis and regeneration remains unclear. The current view posits support cells as multipotent progenitors capable of self-renewing and differentiation and suggests a niche role for mantle cells influencing SC proliferative behaviour and fate (*Romero-Carvajal et al., 2015*). In the absence of long-term lineage tracing data, however, it is very difficult to reveal the identity and location of neuromast stem cells and to assess if homeostatic replacement, response to injuries, and generation of new organs during post-embryonic stages are all performed by the same or different cell types.

Here we use newly developed transgenic lines to follow lineages during development and into adulthood in neuromasts of medaka fish. We prove that mantle cells constitute *bona fide* neuromast neural stem cells during homeostasis, growth and organ regeneration. Additionally, we identify a new population of neuromast cells that we name neuromast border cells (nBCs), which are conserved in other teleost fish. We demonstrate that in medaka, nBCs constitute a different lineage that never crosses boundaries with the neural lineage maintained by mantle cells. We track border cells back to earlier developmental stages both in medaka and zebrafish, and reveal that they do not originate from the pLL primordium but rather from the suprabasal skin epithelium, defining neuromasts as composite organs. Finally, we show that neural stem cells are necessary and sufficient to induce the conversion of epithelial cells into nBCs, and that the ablation of nBCs disrupts the architecture of the organ. Altogether, we uncover that neural stem cells recruit and intimately associate with neighbouring cells that will be maintained as a life-long separate lineage.

## Results

### nBCs are the outer cells of the organ

To address the existence and identity of neuromast stem cells we decided to follow a lineage analysis approach using the *Gaudí* toolkit (*Centanin et al., 2014*), in combination with transgenic lines

that label the different cell types within mature neuromasts. The transgenic line (Tg) Tg(*Eya1*:EGFP) (*Seleit et al., 2017*) is expressed in all cells of the migrating primordium and during organogenesis,

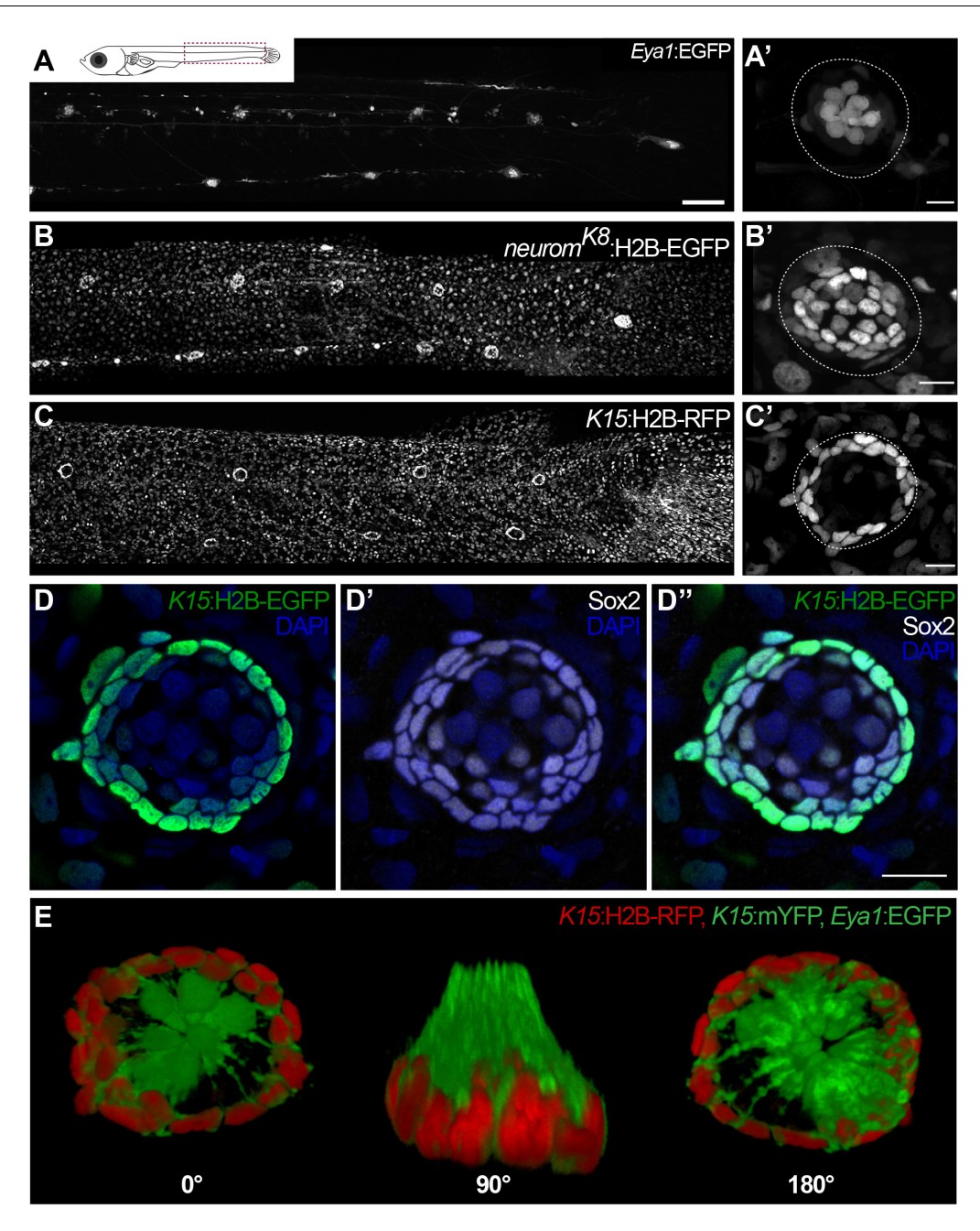

**Figure 1.** Specific transgenic lines label mantle, support and hair cells in mature medaka neuromasts. Tg(*Eya1*:EGFP) allows visualisation of all neuromasts along the pLL (**A**), and labels hair cells and internal support cells (**A'**) (N=>20 neuromasts in >10 larvae). The enhancer trap *neuromK8* line labels skin epithelia (**B**) and mantle and support cells of a mature neuromast (**B'**) (N=>20 neuromasts in >10 larvae). Tg(*K15*:H2B-RFP) also labels skin epithelia all over the body surface (**C**), but RFP expression in mature neuromasts is restricted to mantle cells (**C'**) (N=>10 neuromasts in >10 larvae). *K15*+mantle cells are Sox2 +as revealed by immunostainings (**D–D''**) (N=>10 neuromasts in seven larvae). A 3D reconstruction of a mature neuromast (**E**) of the triple transgenic line Tg(*K15*:mYFP)Tg(*Eya1*:EGFP)Tg(*K15*:H2B-RFP) of an early juvenile. Six neuromast hair cells (green bundles) project outwards, surrounded by a ring of mantle cells (red nuclei and green membranes) that encapsulate the hair cell bundles in a cupula-like structure (N = 6 neuromasts in two larvae). Scalebars are 100 μm for entire trunks (**A**, **B**, **C**) and 10 μm in neuromast close-ups.
DOI: https://doi.org/10.7554/eLife.29173.003

but is restricted to hair cells and internal support cells located underneath HCs in mature organs (*Figure 1A,A'*). The newly generated enhancer trap *neurom^K8*:H2B-EGFP stably labels all support cells and mantle cells (*Figure 1B,B'*)(See Materials and methods). Additionally, the Tg(*K15*:H2B-RFP) labels a subset of the *neurom^K8* positive cells that form a peripheral ring in mature neuromasts (*Figure 1C–C'*) and are Sox2 positive (*Figure 1D–D''*). A 3D reconstruction of triple transgenic Tg (*K15*:mEYFP), Tg(*K15*:H2B-RFP), Tg(*Eya1*:EGFP) neuromasts indicates that EYFP positive cells are wrapping the hair cells of the organ (*Figure 1E* and *Video 1*), a distinctive morphology and position that characterises mantle cells (*Jones and Corwin, 1993*; *Steiner et al., 2014*). Both *neurom^K8*:H2B-EGFP and Tg(*K15*:H2B-RFP) also label skin epithelia on the entire body of juvenile and adult medaka fish. This combination of transgenic lines allows us to dynamically assess the cell content of neuromasts *in vivo* during embryonic, juvenile and adult organ growth.

While observing neuromasts counterstained with DAPI, we noticed that *K15* positive (*K15^+*) mantle cells are consistently surrounded by an outer ring of cells that feature elongated nuclei (*Figure 2A,A'*). This is the case for all neuromasts in medaka, including ventral, midline and dorsal neuromasts on the posterior lateral line, and neuromasts of the anterior lateral lines in both juveniles and adults (N > 100 neuromasts). Since these elongated nuclei locate to the outer border of neuromasts, we named the corresponding cells neuromast Border Cells (nBCs). Electron microscopy revealed that the membranes of border cells are intimately associated with those of mantle cells, often producing cytoplasmic protrusions into one another (*Figure 2B–B'''*) In addition we also observed desmosomes between MCs and nBCs (*Figure 2C–C''*). Using iterative imaging on Tg(*K15*:H2B-EGFP) medaka larvae, we detected that a proportion of nBCs are EGFP positive (*Figure 2D*) but this expression decays within the following days. The transient nature of this expression suggests nBCs could originate from *K15^+* cells and inherit the fluorescent protein, which in this case would be acting as a short-term lineage tracer. We therefore focused on revealing the embryonic origin and lineage relations of all neuromast cell types (*Figure 2E*) during homeostatic maintenance, organ growth and post-embryonic organogenesis.

## nBCs constitute an independent life-long lineage

To understand the lineage relations between the different cell types of mature neuromasts, we labelled individual cells and followed clones over time using the lineage-tracing Gaudí toolkit (*Centanin et al., 2014*). Briefly, the Gaudi toolkit consists of driver Cre recombinase lines and reporter LoxP lines that, when crossed to each other, allow labelling a cell and following its entire progeny life-long by the expression of a fluorescent protein that is absent in non-recombined cells (*Figure 3A*). We generated clones by inducing sparse recombination in *Gaudí^RSG* (*ubiquitin*:LoxP-DsRed-LoxP-H2B-EGFP) crossed to either *Gaudí^Ubiq.iCre* (*ubiquitin*:Cre^ERT2) or *Gaudí^Hsp70.A* (*Hsp70*:nlsCre) embryos and identified cell types based on their relative position and cell size (*Figure 3—figure supplement 1*). We imaged neuromasts 2 days post induction (*Figure 3B*), and decided to follow the ones containing up to three clones, a clone being defined as 1–4 adjacent cells (*Figure 3—figure supplement 2*). The lineage analysis of 152 clones in 111 neuromasts (21 embryos) from 9 and up to

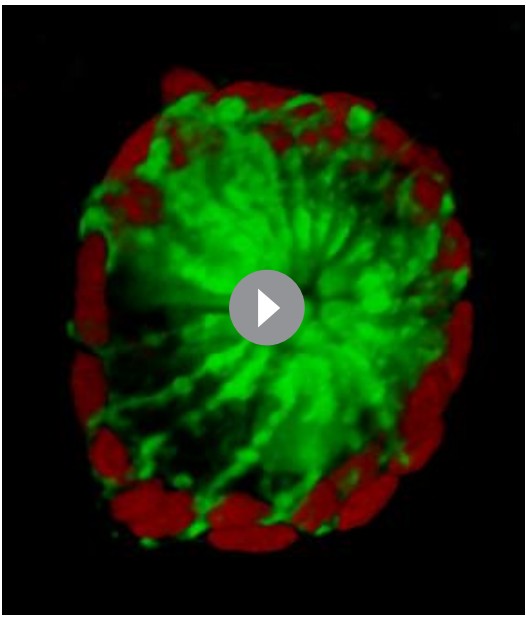

**Video 1.** 3D reconstruction of cell types of mature neuromasts. Location and orientation of cells within the neural lineage of a mature neuromast organ in the triple transgenic line Tg(*K15*:mYFP) (*K15*:H2BRFP)(*Eya1*: GFP) in a stage 42 medaka embryo. Six neuromast hair cells (green bundles) labelled with *Eya1*:GFP are projecting outwards into the environment, they are surrounded by an outer ring of mantle cells (red) labelled with *K15*:H2B-RFP. Mantle cells bodies encapsulate the hair cell bundles in a cupula-like structure (green membranes) labelled with *K15*:mYFP (N = 6 neuromasts in two larvae).
DOI: https://doi.org/10.7554/eLife.29173.002

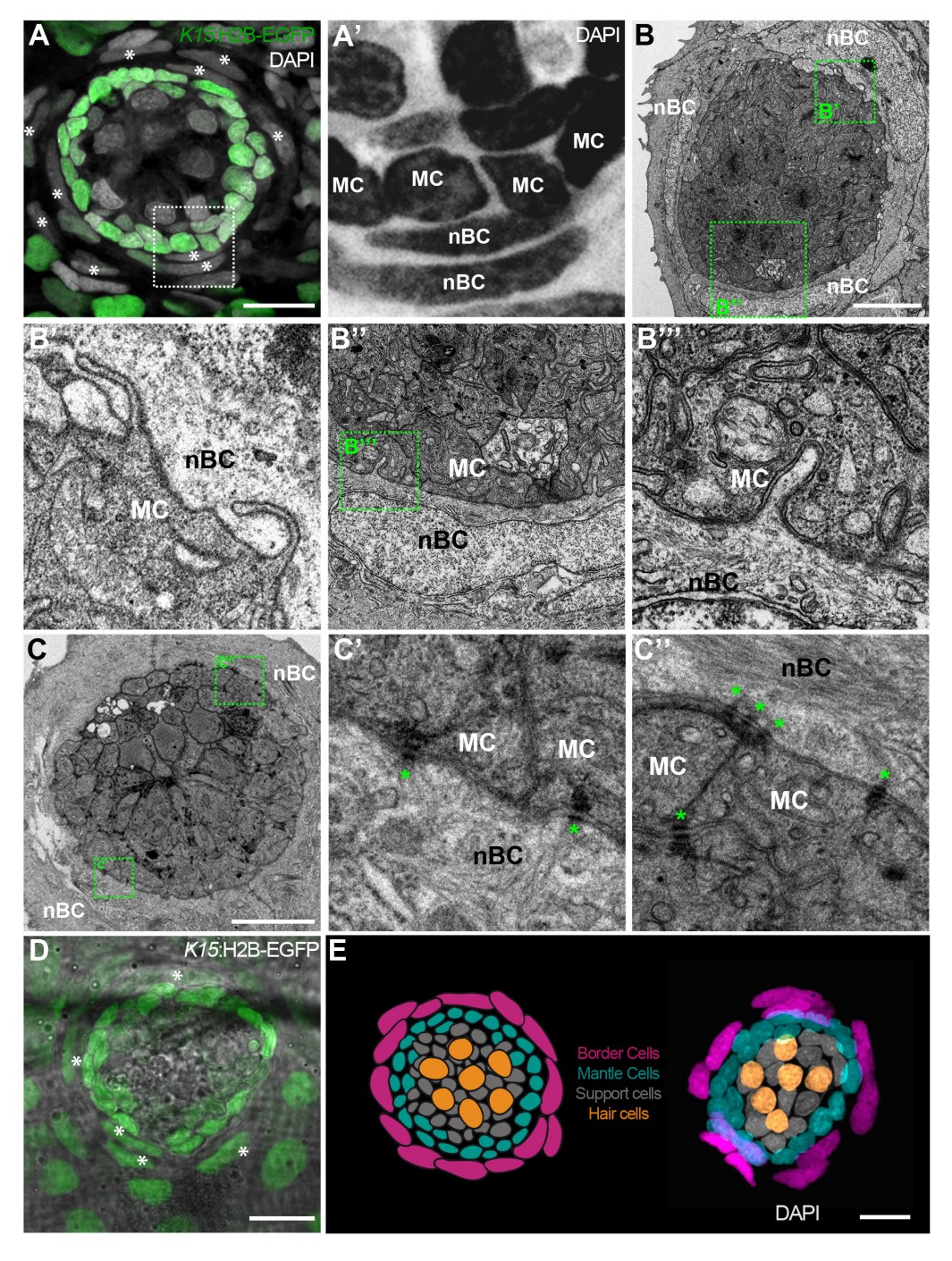

**Figure 2.** nBCs surround mantle cells of the neural lineage in mature neuromasts. Early juvenile neuromasts from Tg(K15:H2B-GFP) show mantle cells (green in A, 'MC' in A') that are closely surrounded by elongated nuclei (border cells, nBCs) visualised by DAPI (white asterisks in A, 'nBC' in A') (N=>10 neuromasts in >5 larvae). DAPI is shown in grey (A) and black (A') to enhance contrast. Electronmicroscopy reveals that nBC and K15 +mantle cells are in close contact (B–C'') (N = 6 neuromasts in four fish). (B) Overview of a mature neuromast where mantle cells are surrounded by three nBCs. (B'–B''') Zoom-in panels from figure (B) reveal a close association between MCs and nBCs that includes cytoplasmic protrusions of each cell type into the other (B'–B''') (N = 6 neuromasts in four fish). (C) An upper section on a neuromast where mantle cells are surrounded by two nBCs. The darker dots (green asterisks) are cytoplasmic plaques of desmosomes formed between mantle cells and nBCs (C', C'') (N = 2 neuromasts in one fish). (D) A younger neuromast than the one depicted in (A) from Tg(K15:H2B-GFP) shows that nBCs are also labelled with GFP (white asterisks in D) (N=>10 neuromasts in >5 larvae). (E) Pseudo-coloured DAPI neuromast and scheme depicting the four cell types observed in every mature neuromast organ. Hair cells are shown in yellow, support cells in grey, mantle cells in green and border cells in magenta. Scalebars are 10 μm.

DOI: https://doi.org/10.7554/eLife.29173.004

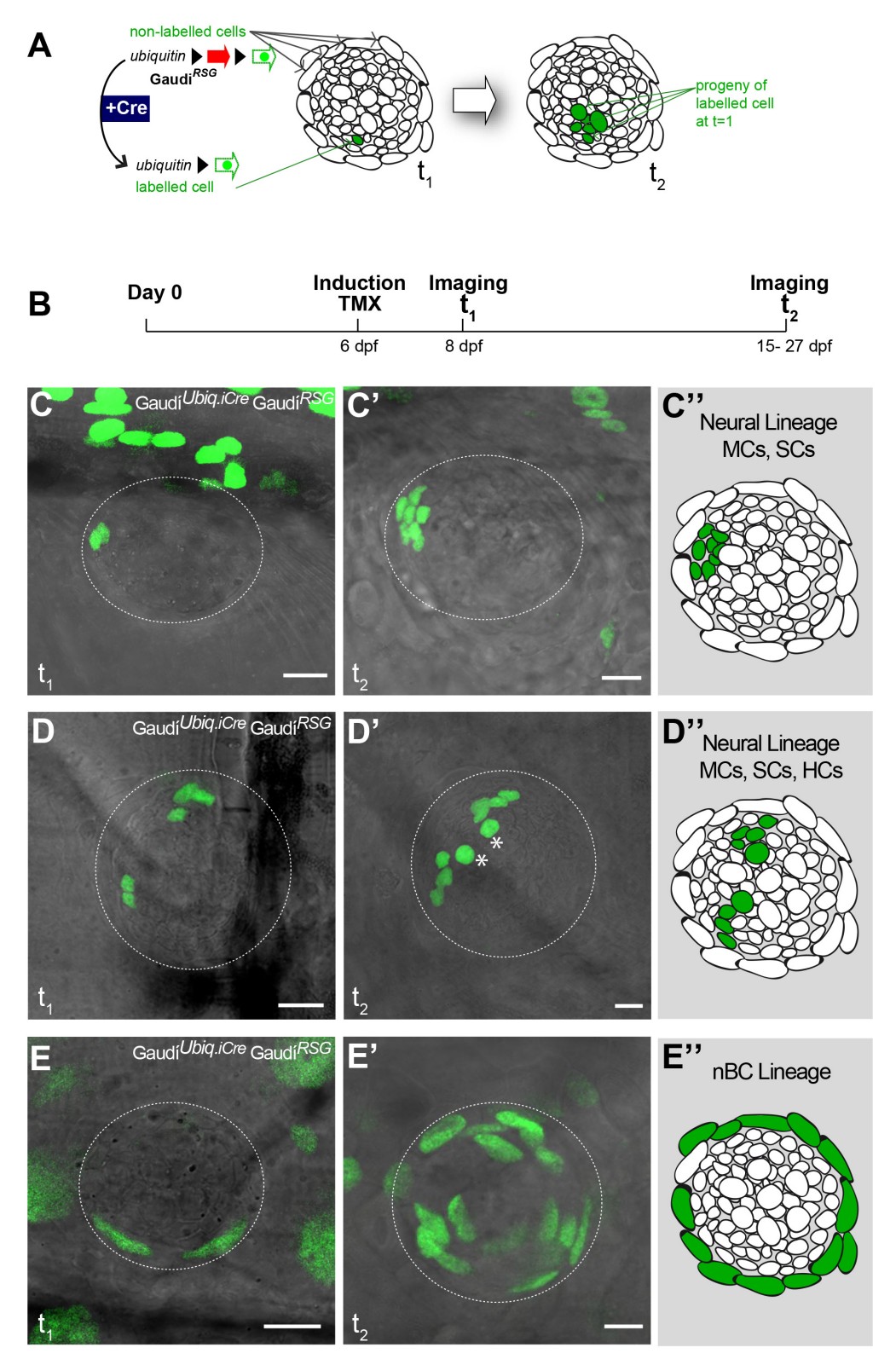

**Figure 3.** Mature neuromasts are composed of two separate lineages. (**A**) Experimental outline of clonal induction and short-term lineage tracing. t2 = 15 dpf. for B' and D', and 27 dpf for C'. (**B–D''**) Iterative imaging of the same clones (H2B-EGFP) after recombination via tamoxifen reveals different behaviour of neuromast cells. Live imaging of clones containing mantle cells at t1, which expanded into the support cell domain (B,B' and scheme in B'') and even further to generate hair cells (C, C' and scheme in C'). Clones that contain labelled nBCs at t1 were restricted to the nBCs domain later on

*Figure 3 continued on next page*

*Figure 3 continued*

(D, D' and scheme in D'), revealing a separate lineage for MCs-SCs-HCs and nBCs. Scalebars are 10 μm. ($N_{total}$ = 152 clones in 111 neuromasts of 21 embryos).

DOI: https://doi.org/10.7554/eLife.29173.005

The following figure supplements are available for figure 3:

**Figure supplement 1.** Bright-field imaging allows defining the boundary between nBCs and neural lineage.

DOI: https://doi.org/10.7554/eLife.29173.006

**Figure supplement 2.** Examples of different neuromast clones induced by the Gaudi[RSG] lineage tracing tool.

DOI: https://doi.org/10.7554/eLife.29173.007

21 days suggested two independent short-term lineages. One lineage involves mantle cells that expand and generate support and hair cells (*Figure 3C–D''*) - the neural lineage - while the other lineage was restricted to neuromast border cells (*Figure 3E–E''*). We observed that 72 neuromasts contained clones that were restricted to the neural lineage throughout the timeline of the experiment. Intriguingly, they never generated a labelled nBC, suggesting that border cells do not originate from this lineage. These 72 neuromasts included cases of the entire proposed neural lineage progression from mantle to support to hair cells (N = 7 clones in 7 neuromasts in four fish) (*Figure 3D–D''*). We also observed 21 neuromasts containing labelled nBCs, which did not contribute to the neural lineage (*Figure 3E–E'*). The remaining 18 neuromasts contained clones in both lineages, which could be explained either by separate, distinct labelling events or by the presence of a bi-potent stem cell.

To explore whether this fate-restriction is maintained life-long, we focused on the *caudal neuromast cluster* (CNC) because of its stereotypic location on the caudal fin (*Wada et al., 2008*). The CNC contains an increasing amount of neuromasts as fish age - the older the fish, the more neuromasts in the CNC. Neuromasts in the CNC are generated post-embryonically, presumably from a founder, embryonic neuromast - neuromast[P0] (*Figure 4A,B*). Neuromast[P0] is the last neuromast formed by the migrating primordium as it comes to a halt and takes its position in a highly stereotypic way on what will become part of the caudal fin. We confirmed that it is the source of all new organs in the CNC by two photon laser ablations, which revealed that eliminating the neuromast[P0] at eight days post-fertilisation (dpf)(*Figure 4C,C'*) results in an adult missing the CNC on the experimental side but with a wild-type CNC on the contralateral, control side (*Figure 4D,E*). The CNC represents a system that allows us to investigate neuromast stem cells during organ growth, homeostasis and also during post-embryonic organ formation.

We induced recombination in late embryos that were grown for two and up to 18 months post induction (*Figure 5A*). The analysis of long-term clones in the CNC revealed that both neural and nBC lineages are maintained by dedicated, fate-restricted stem cells (*Figure 5B–D*). We focused our analysis on CNCs in which 75% or less neuromasts contained labelled cells (47 CNCs containing 284 neuromasts, from which 112 contained GFP positive cells, in 30 recombined Gaudi[RSG] fish). We observed that 92% of neuromasts contain fate restricted clones, comprising either the neural lineage (34.8%, N = 39/112 neuromasts)(*Figure 5B,D*) or the nBC lineage (57.1%, N = 64/112 neuromasts) (*Figure 5C,D*). The remaining 8% of neuromasts show co-occurrence of labelled cells in both lineages (N = 9/112 neuromasts)(*Figure 5—figure supplement 1*). These co-labelled cases could be the result of simultaneous recombination of cells in both lineages or alternatively, be produced by a rare bi-potent stem cell. By restricting the analysis to CNCs with sparse labelling ($\leq$50% of neuromasts labelled per cluster), the ratio of co-labelled clones drops further (5.1%, 4/79 labelled neuromasts in 39 CNCs), and disappears when we select for an even lower recombination efficiency ($\leq$25% of neuromasts labelled per cluster, 0/17 labelled neuromasts in 15 CNCs). Although we cannot exclude the existence of a bi-potent stem cell, the presence of fate-restricted stem cells is further supported by CNCs in which all neuromasts are labelled in either the nBC (N = 53 neuromasts in 10 CNCs) or the neural lineage (N = 3 neuromasts in 1 CNC) (*Figure 5—figure supplement 2*). Taken together, our results indicate the existence of independent lineages during neuromast homeostasis and post-embryonic organogenesis, and position the neuromast as a minimal system to tackle stem cell fate-restriction and clonal organisation in a composite organ.

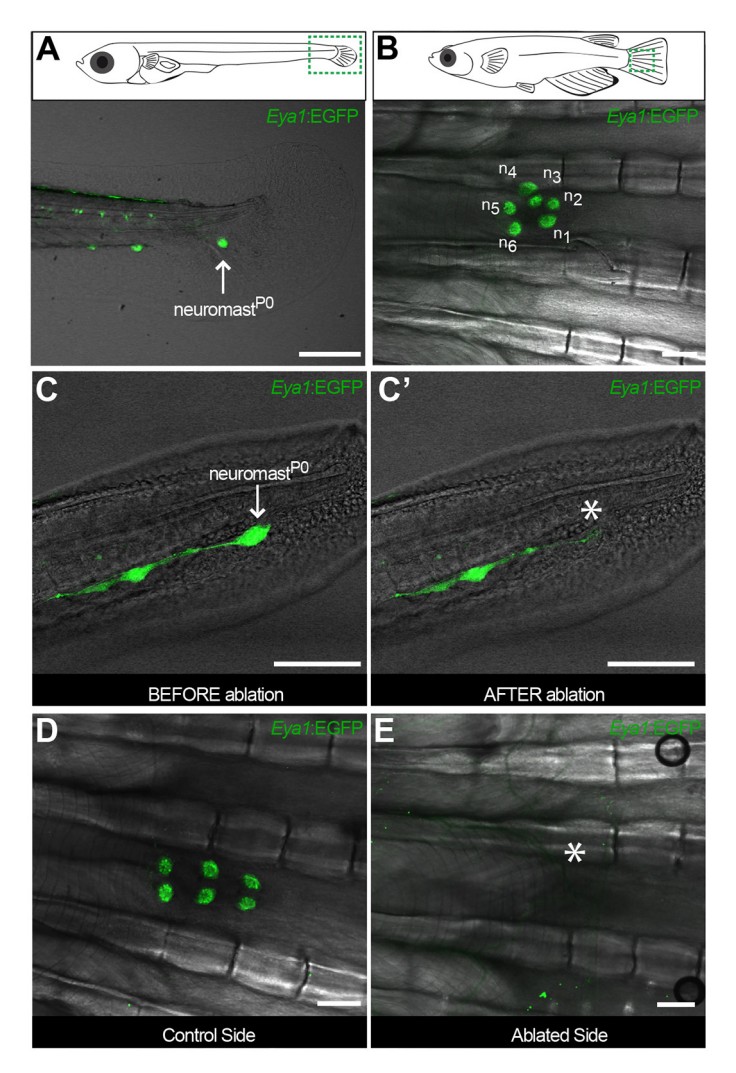

**Figure 4.** The CNC allows studying stem cells during organ growth, homeostasis and post-embryonic organ formation. Early juvenile medaka display a single neuromast on the caudal fin (neuromast$^{P0}$, arrow in A), which gives rise to a cluster of neuromasts (n1 to n6, (**B**) during post-embryonic life. Two-photon laser microscopy can be used to effectively ablate neuromast$^{P0}$ (**C** and asterisk in **C'**). The caudal neuromast cluster (**D**) cannot form in the absence of neuromast$^{P0}$ (asterisk in **E**) (n = 7 CNCs in seven fish). Scalebars are 10 μm in embryos and 100 μm in adults.

DOI: https://doi.org/10.7554/eLife.29173.008

## Mantle cells are neural stem cells

Having shown that neuromasts contain stem cells that maintain the neural lineage, we tackled neuromast stem cell identity using a regeneration approach. Combining Tg(*K15*:H2B-RFP) with Tg(*Eya1*:EGFP) or Tg(*Eya1*:H2BGFP), we ablated a major proportion (40% up to 95%) of neural lineage cells using two-photon laser confocal microscopy sparing a few intact $K15^+$ mantle cells (N = 18 neuromasts in four fish)(*Figure 6A–B'*). Iterative post-injury imaging revealed that the surviving mantle cells initially coalesce and then at 40 hours post injury increase their numbers resulting in a small circular cluster of *K15* +cells (*Figure 6C–C''*). This regenerating neuromast progressively increases in size and eventually shows *Eya1* positive internal SCs and HCs (*Figure 6D–D''*). Notably, as few as four $K15^+$ cells were sufficient to regenerate the entire neural lineage of a neuromast. In addition we performed similar ablations in the double Tg(*K15*:H2B-RFP)(*neurom*$^{K8}$:H2B-EGFP) in which all support cells and a proportion of mantle cells were ablated that resulted in completely regenerated

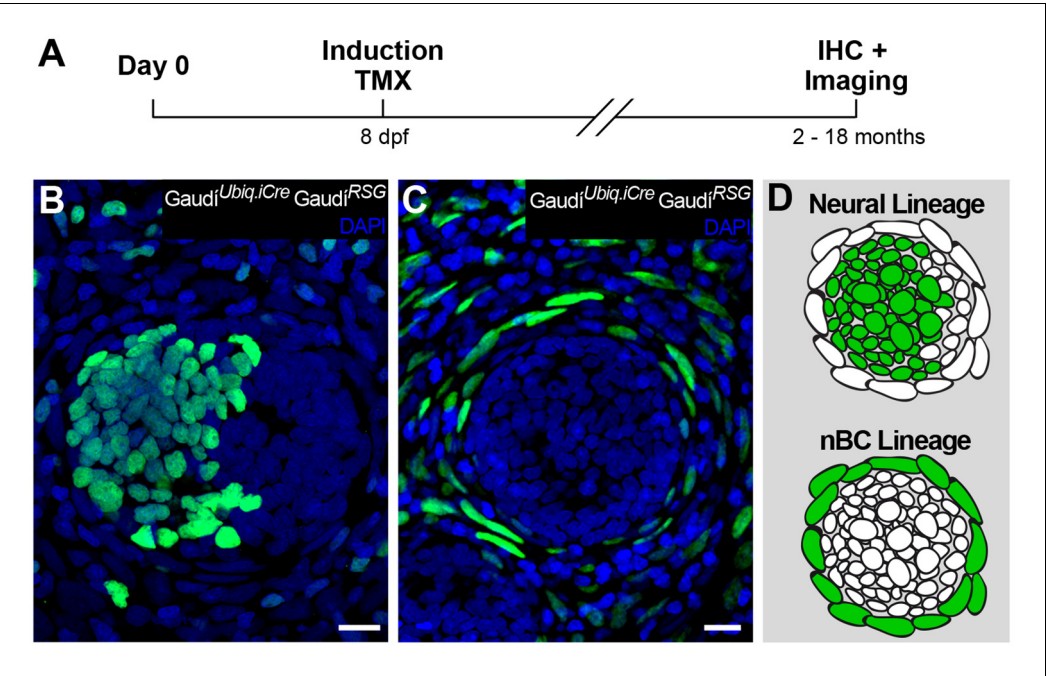

**Figure 5.** Independent stem cell populations maintain neural and nBC lineages in mature neuromasts. (**A**) Experimental outline of long-term lineage tracing using Gaudi$^{RSG}$ Gaudi$^{Ubiq.iCre}$. Neuromasts in the CNC of induced fish are labelled either in the neural lineage (B, upper scheme in D)(n = 39 neuromasts in 47 CNCs) or in the nBC lineage (C, bottom scheme in D) (n = 64 neuromasts in 47 CNCs). Neuromasts labelled in the neural lineage contain mantle, support and hair cells (**B**), while clones in the nBC lineage do not contribute to the neural lineage (**C**). Scalebars are 10 μm.

DOI: https://doi.org/10.7554/eLife.29173.009

The following figure supplements are available for figure 5:

**Figure supplement 1.** High labelling efficiencies result in neuromast labelled in both neural and nBC lineages.
DOI: https://doi.org/10.7554/eLife.29173.010

**Figure supplement 2.** Independent lineages occur even in highly recombined CNCs.
DOI: https://doi.org/10.7554/eLife.29173.011

organs that contained all three neural lineage cell types (N = 8 neuromasts in four larvae). These results indicate that mantle cells have the potential of reconstituting all the cell types within the neural lineage of an organ after severe injury.

To assess whether $K15^+$ mantle cells function as neural stem cells during homeostasis and post-embryonic organogenesis, we generated Tg($K15$:Cre$^{ERT2}$) to permanently label mantle cells and their progeny. When crossed to *Gaudi$^{RSG}$* and induced for recombination, the larvae exhibited sparse recombination in skin epithelial cells and in mantle cells as expected from the Tg($K15$:H2B-EGFP) (*Figure 7A–C*). We imaged $K15^+$ recombined cells at 72 hr post-induction, selecting for clones containing few mantle cells (*Figure 7D,E*). These selected fish were grown and clones were imaged again up to one month later. Our analysis revealed that mantle cell derived clones contained mantle and support cells (N = 30 neuromasts in 12 fish) (*Figure 7D'*) and in some cases all cell types of the neural lineage (*Figure 7E'*) (N = 8 neuromasts in five fish).

Additionally, we induced Tg($K15$:Cre$^{ERT2}$) Gaudi$^{RSG}$ embryos and grew them to adulthood to examine K15-derived clones in the CNC. In line with our previous results, the analysis of long-term clones in the CNC further supported independent cells of origin for neural and nBC lineages (*Figure 7F–G*)(15 CNCs containing 59 neuromasts, from which 32 contained GFP$^+$ cells, in 14 fish). We observed fate restriction in 100% of cases analysed (32/32 labelled neuromasts), comprising either the neural lineage (87.5%, 28/32 labelled neuromast) or the nBC lineage (12.5%, 4/32 labelled neuromasts). We were able to follow a clone of three mantle cells from embryonic neuromast$^{P0}$ to an 11 month old CNC that contained other neuromasts with EGFP$^+$ cells in the entire neural lineage

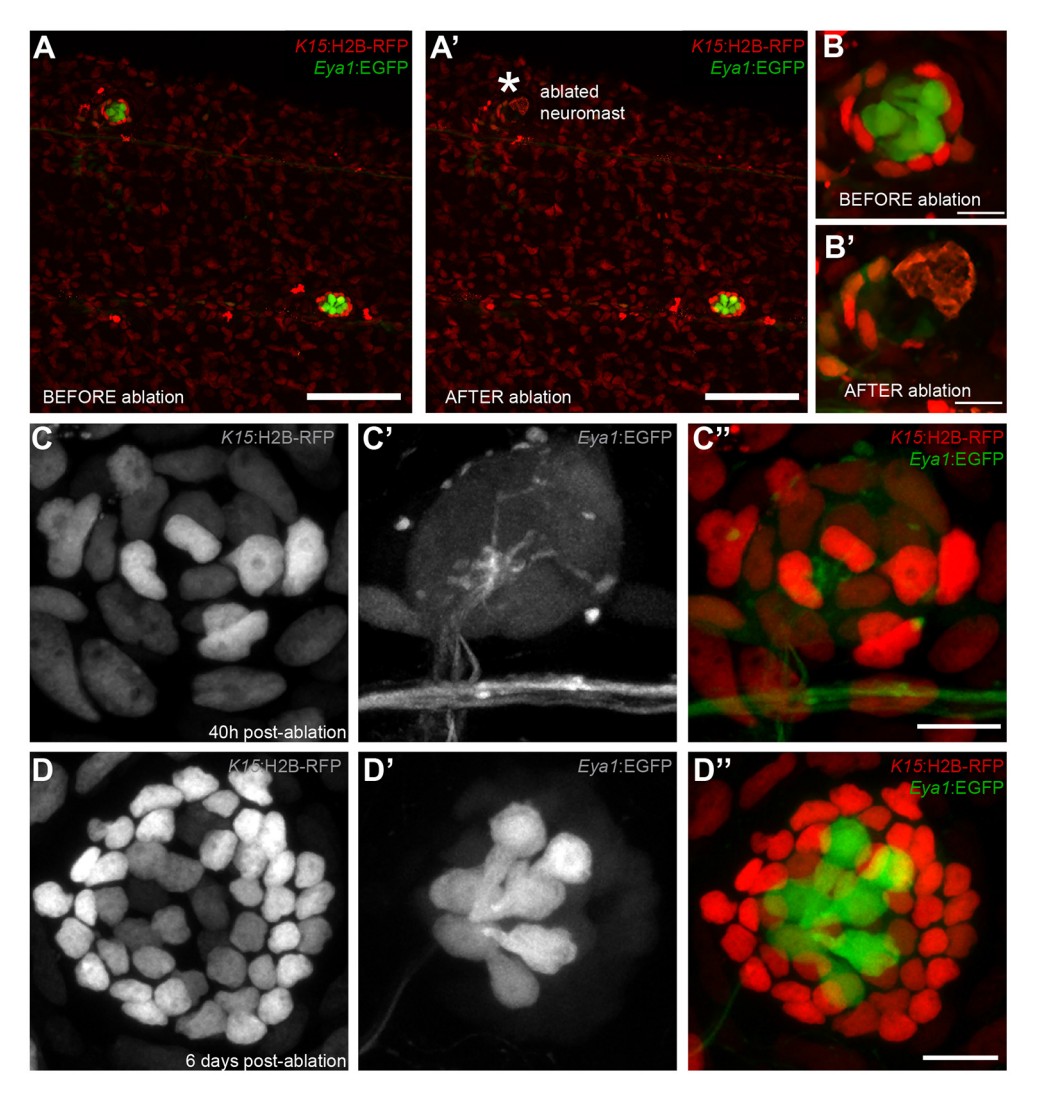

**Figure 6.** Mantle cells regenerate all cell types within the neural lineage upon severe injury. Two photon ablation on neuromasts of the double transgenic Tg(*K15*:H2B-RFP), Tg(*Eya1*:EGFP) at 12 dpf. (**A–B'**). Ablations were done to remove most cells in the neuromasts, sparing a few *K15*+ cells (**B'**). The same neuromast shown 40 hr post-ablation reveals a small cluster of RFP +cells that have coalesced around the site of injury without any apparent differentiation (**C–C''**). Six days post-injury all cell types within the neural lineage have been reconstituted (**D–D''**). Mantle and support cells can be observed in *K15*:H2B-RFP (**D, D''**), while hair cells and internal support cells are evident in the *Eya1*:EGFP (**D', D''**). (N = 18 neuromasts in four larvae for Tg(*K15*:H2B-RFP), Tg(*Eya1*:EGFP)), N = 8 neuromasts in four larvae for Tg(*K15*:H2B-RFP)(*neurom*^K8:H2B-EGFP)) Scalebars are 100 μm for trunks (**A, A'**) and 10 μm for neuromasts (**B–D''**).

DOI: https://doi.org/10.7554/eLife.29173.012

(3 out of 5 neuromasts in the CNC)(*Figure 7H', H''*). This indicates that *K15*+ mantle cells also function as founder cells for post-embryonic organogenesis. Altogether, our results indicate that *K15*+ mantle cells function as neuromast neural stem cells during homeostatic growth, post-embryonic formation of new neuromasts and organ regeneration.

## Independent embryonic origins for border and neural lineage

The presence of labelled nBC clones in Tg(*K15*:Cre^ERT2) Gaudi^RSG was also observed in neuromasts of the midline pLL (N = 5 neuromasts in three fish). This is compatible with the transient expression of EGFP we had previously observed in Tg(*K15*:H2B-EGFP) hatchlings, and suggests that nBCs

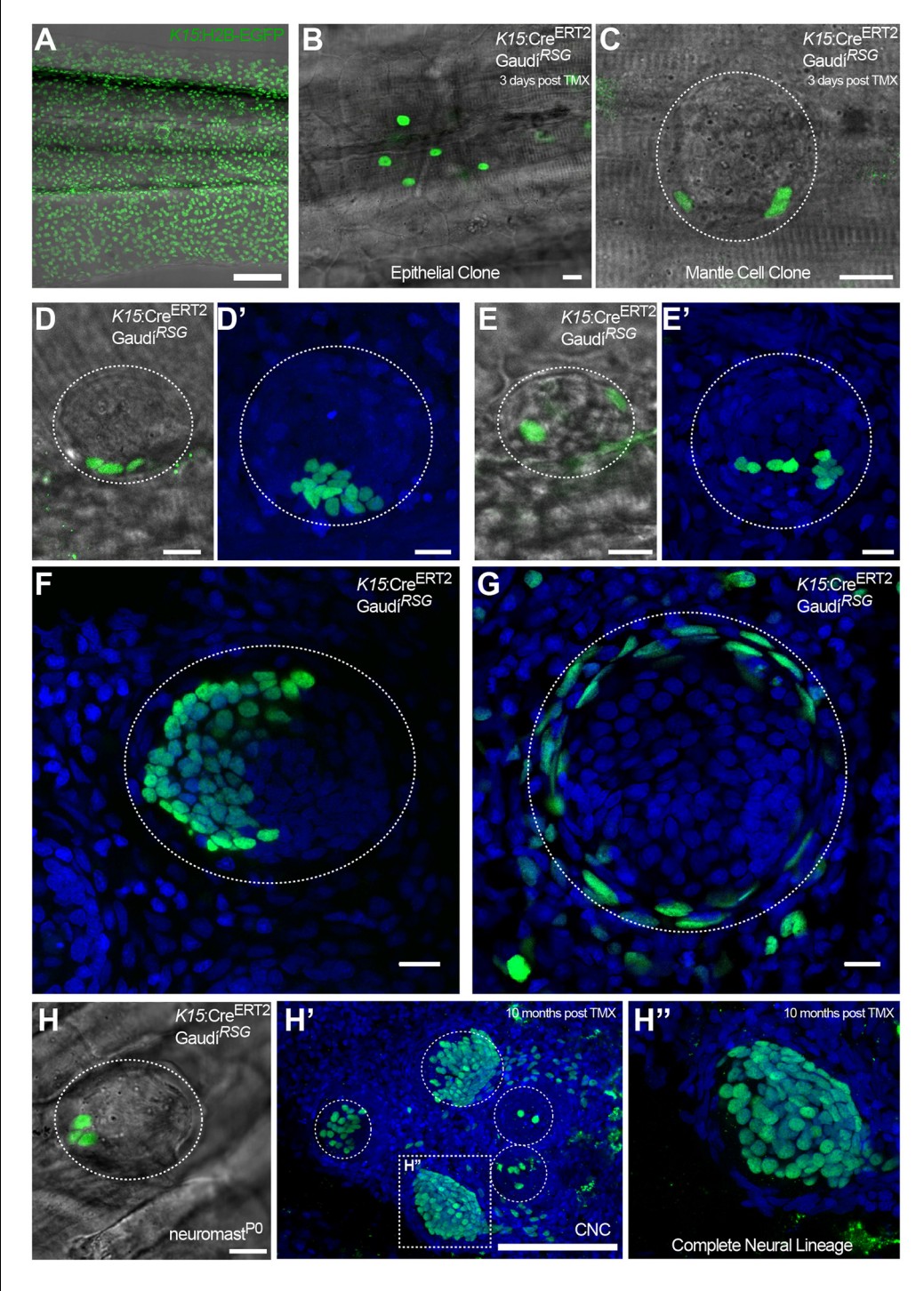

**Figure 7.** *K15⁺* mantle cells are neuromast neural stem cells. Tamoxifen induction of *K15*:Cre^ERT2^, *Gaudi*^RSG^ labels a subset of the EGFP⁺ cells of Tg(*K15*:H2B-EGFP) in the skin epithelium and the neuromast (Compare **A** to **B**, **C**). A clone of three mantle cells in induced *K15*:Cre^ERT2^, *Gaudi*^RSG^ (**D**) generates more mantle and also support cells (**D'**). A second clone of labelled mantle cells (**E**) expanded into support and differentiated hair cells (**E'**). Adult neuromast showing K15 derived clones in the neural (**F**) and in the nBC (**G**) lineages. A clone of three mantle cells in *neuromast*^P0^ (**H**) contributes to 3 different neuromasts in the adult caudal neuromast cluster (**H'**) and generate all cell types of the neural lineage (**H''**) (N = three neuromast in one fish). Wide scalebars are 100 μm for trunks (**A**, **F'**) and thinner scalebars are 10 μm.

DOI: https://doi.org/10.7554/eLife.29173.013

*Figure 7 continued on next page*

*Figure 7 continued*

The following figure supplement is available for figure 7:

**Figure supplement 1.** Independent origin of neural and nBC lineages in medaka neuromasts.

DOI: https://doi.org/10.7554/eLife.29173.014

indeed derive from *K15*⁺ cells. The fact that the same marker labels fate-restricted stem cells for two different lineages can be explained by two possible scenarios. Neuromast BCs could originate either from a subset of *K15*⁺ mantle cells that are committed to exclusively produce border cells, or from *K15*⁺ cells outside of the neuromast neural lineage.

To address nBC origin, we generated mosaic embryos by injecting a *K15*:H2B-EGFP plasmid at the 2–4 cell stage into wild types. Injected embryos at 9 dpf were stained with antibodies against EGFP to allow a longer time window analysing the progeny of *K15*⁺ cells. This revealed neuromasts that were either labelled in the neural lineage (N = 15/37 neuromasts in seven fish) or in the nBC lineage (N = 11/37 neuromasts in seven fish) suggesting that the two lineages are already set apart by the time of organ formation (*Figure 7—figure supplement 1A,B*). Expectedly, we also observed neuromasts in which both border cells and the neural lineage were labelled (N = 11/37 neuromasts in seven fish) mirroring our observations with long term lineage analysis. To follow *K15*⁺ progeny for longer periods, we injected a *K15*::Lex^PR^ Lex^OP^:CRE plasmid into *Gaudi^RSG^* since the LexPR system provides an amplification step (*Emelyanov and Parinov, 2008*; *Lust et al., 2016*) that results in higher recombination rates than the *K15*:Cre^ERT2^ approach (see Materials and methods). Here again, we observed neuromasts that were either labelled in the neural lineage (N = 3/13 neuromasts) or in the border cell lineage (N = 9/13 neuromasts) (*Figure 7—figure supplement 1C,D*). Notably, we observed that clones containing border cells reproducibly continue into skin epithelial cells in the vicinity of the neuromasts both in injected embryos (100%, N = 40 neuromasts)(*Figure 7—figure supplement 1B,D*) as well as in the previously described long term lineage analysis (100%, N > 100 neuromasts)(*Figure 5—figure supplement 2*, *Figure 5C*). Taken together, these results indicate that nBCs could originate from *K15*⁺ cells within the epithelium.

## nBCs originate from K15⁺ cells in the epithelium

To dynamically address the developmental origin of border cells, we followed a 4D approach using Tg (*K15*:H2B-EGFP) embryos. We exploited the migration of the developing midline neuromasts to their final destination (*Seleit et al., 2017*) to follow the two *K15*⁺ populations during organogenesis (*Figure 8A* and *Videos 2–4*). As the developing midline neuromasts reach the horizontal myoseptum they come into contact with the overlying epithelial cells. Promptly, three to four *K15*⁺ epithelial cells (red, green and blue dots in *Figure 8A*) respond to the arrival of the neural stem cell precursors. Over a period of 72 hours, these epithelial cells undergo significant morphological changes resulting in the characteristic elongated shape of border cell nuclei. Both ventral and midline neuromasts of the posterior lateral line system in medaka, as well as neuromasts of the anterior lateral line (aLL), trigger the same behaviour in the surrounding epithelium (five ventral neuromasts and four midline neuromasts in the pLL and two neuromasts in the aLL, N = 5 embryos). These movies indicate that the arrival of neural stem cell precursors is accompanied by a change in nuclear morphology of skin epithelial cells that were present at the place of arrival.

We wondered whether in addition to a change in nuclear shape, the cellular membranes of skin epithelial cells also undergo morphological remodelling, so we performed double injections of *K15*:H2B-RFP and *K15*:mYFP plasmids into *Cab* wild type embryos to create clones labelled for nuclei and membranes. The results revealed that a change in cellular morphology coincides with a change in nuclear shape (*Figure 8B,B'*, white and green asterisks, *Video 5* and quantifications in *Figure 8—figure supplement 1*), as observed in clones that contain both skin epithelial cells and nBCs. A detailed analysis of nBC clones revealed that these are continuous with the suprabasal layer of the skin epithelium rather than with more superficial epithelial cells (see *Figure 8—figure supplement 2* and *Video 5*). In addition, the injection of the *K15*:mYFP plasmid allowed us to analyse the membranes of neural stem cell precursors prior to and during the induction phase. We observed that *K15*⁺ cells of the neural lineage extended numerous cytoplasmic processes during the early phase of organ formation (*Figure 8—figure supplement 3*) (N = 8 clones in five larvae). Iterative imaging of

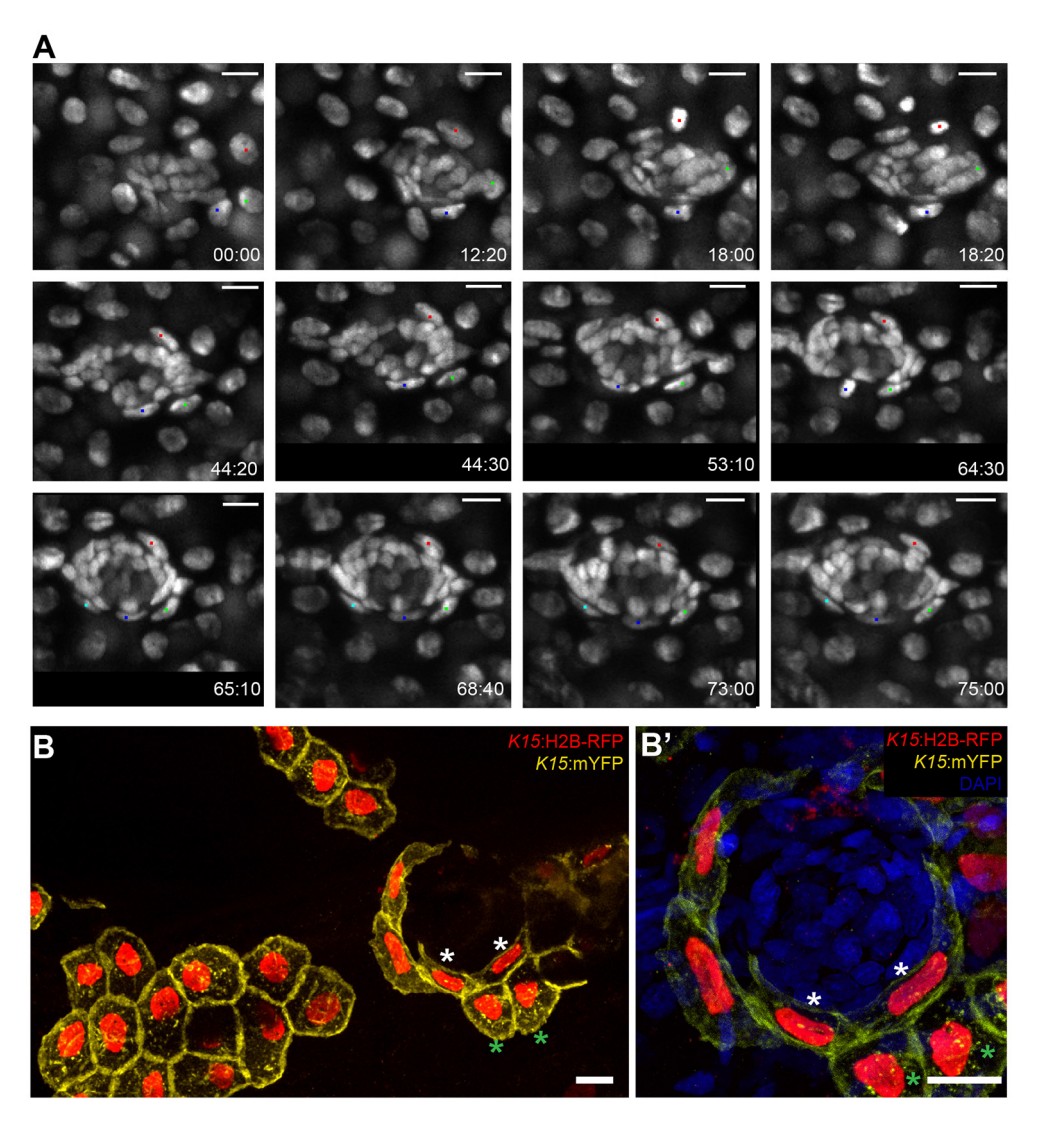

**Figure 8.** Induction of skin epithelial cells into nBCs during neuromast formation. (**A**) Time-lapse imaging of a stage 35 Tg(*K15*:H2B-EGFP) embryo during secondary organ formation, where three epithelial cells (blue, green and red dots) dynamically associate with the arriving neural stem cells precursors. The red cell divides (18:00 hr to 18:20 hr) to generate one skin epithelial daughter and another daughter that will become a nBC (64:00 hr to 75:00 hr). The green cell transitions into a nBC without dividing, and the blue cell first becomes a nBC (18:20 hr to 43:00 hr) and then divides to generate two nBCs (64:30 hr to 65:10 hr) that stay in the neuromast. The images in (**A**) are selected time-points from three consecutive movies (see *Videos 2–4*) of the same developing neuromast (five ventral neuromasts and four midline neuromasts in the pLL and two neuromasts in the aLL, N = 5 embryos). (**B–B'**) Immunostaining of a double injected (mosaic) *K15*:mYFP, *K15*:H2B-RFP embryo shows that nBC induction involves a drastic remodelling of both nuclear and cellular morphologies (**B**) (N=12 neuromasts in six larvae). Compare skin epithelial cells (green asterisks in **B, B'**) with their sibling nBCs (white asterisks in B,B') included in the neuromast. Scalebars are 10 µm.

DOI: https://doi.org/10.7554/eLife.29173.018

The following figure supplements are available for figure 8:

**Figure supplement 1.** Changes in shape of cell nuclei correlate with changes in cell morphology.
DOI: https://doi.org/10.7554/eLife.29173.019

**Figure supplement 2.** nBCs are continious with the suprabasal skin epithelium.
DOI: https://doi.org/10.7554/eLife.29173.020

**Figure supplement 3.** Cytoplasmic protrusions from neural stem cell precursors during organ formation

*Figure 8 continued*

DOI: https://doi.org/10.7554/eLife.29173.021

$K15^+$ clones revealed that these processes were dynamic and transient (*Figure 8—figure supplement 3C–C''*), and eventually disappeared by the time of organ maturation. Taken together, these results indicate that nBCs are recruited from $K15^+$ suprabasal skin epithelial cells and reveal numerous cytoplasmic processes from the neural lineage during the induction phase.

## nBCs are a physical niche maintaining organ architecture

So far, we have shown that neural stem cell precursors induce and intimately associate with nBCs during neuromast formation, and form a stable organ architecture that is maintained life-long. Since nBCs form a significant part of the direct $K15+$ mantle stem cell microenvironment, we reasoned that they might constitute a niche for neuromast neural stem cells. We therefore investigated the presence of nBCs in neuromasts of other teleost fish; the closely related amazon molly *Poecilia formosa* and the evolutionarily distant zebrafish, *Danio rerio* – which diverged from medaka ca. 250 millon years ago (*Schartl et al., 2013*). Neuromasts from both fish displayed nuclei that are highly reminiscent of nBCs and that were adjacent to cells with the nuclear morphology of mantle cells (*Figure 9A–D*) (N = 10 neuromasts in a single 1 month old *Poecilia formosa* fish, N = 5 neuromasts in three zebrafish larvae). To better characterise nBCs in zebrafish, we generated a Tg(*o.l.K15*:H2B-EGFP) using the medaka $K15$ promoter sequence and observed expression in the skin epithelium and neuromast mantle cells of 5 days post fertilization larvae (*Figure 9E*) (N = 5 neuromasts in three larvae). Furthermore, and as shown for medaka (*Figure 2D*), zebrafish nBCs transiently expressed low levels of H2B-EGFP suggesting they could also originate from skin epithelial cells (*Figure 9F*). To confirm their embryonic origin in *Danio rerio*, we followed neuromast organogenesis in vivo using a double Tg(*cxcr4b*:Cxcr4b-EGFP)(*o.l.K15*:H2B-EGFP)(*Video 6*). These movies show that despite the different way in which neuromasts are formed in both fish models (*Seleit et al., 2017*), zebrafish nBCs are also induced from the skin epithelium soon after the deposition of the neural lineage by the migrating primordium (N = 4 neuromasts in two embryos). Overall, these results indicate that both the existence and induction of nBCs are conserved in distantly related teleost fish.

While conservation in different fish would suggest nBCs are necessary for mature organs, it does not prove a functional role for these cells in neuromasts. We therefore attempted to experimentally tackle this by ablating nBCs in mature organs and analysed the effect on the neural lineage. The ablation of single nBCs in Tg(*K15*:H2B-EGFP) results in an immediate reaction of the directly adjacent mantle stem cells (*Figure 10A–A'* and *Video 6*). This involves a retraction of mantle cells towards the centre of the neuromast leading to a notch-shaped organ with a distorted architecture (N = 10 neuromast distributed in seven medaka embryos) (*Figure 10A'', A''' Video 7*), suggesting they were detached from their physical anchor. The effect is more drastic when most nBCs surrounding an immature neuromast are ablated in the double Tg(*K15*:H2B-ERFP)(*neurom^K8*:H2B-EGFP) (*Figure 10B,B'*). To explore whether the $K15^+$ mantle stem cells would lose their stem cell marker or undergo apoptosis, we iteratively imaged the neuromasts after injury. We

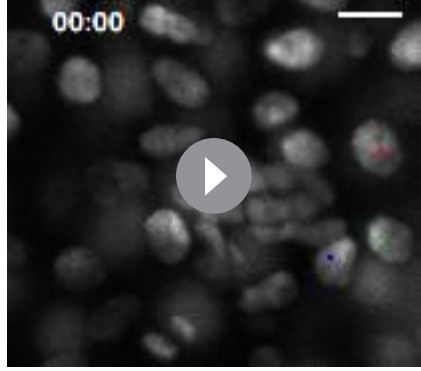

**Video 2.** Induction of epithelial cells into nBCs (1). Time-lapse SpiM imaging of a stage 33–35 Tg(*K15*:H2B-EGFP) embryo during secondary organ formation reveals the developmental origin of nBCs. Coalesced neural stem cell precursors (rosettes) from the posterior lateral line primordium are migrating to their final position. As they do they come into contact with the overlying skin epithelial cells (red, blue and green dots) and a tight association between both cell types signals the beginning of the induction of epithelial cells into nBCs. The length of the movie is 44 hr and 20 min. Stacks were taken every 20 min. Scale bar = 10 μm. Imaging was performed at room temperature.
DOI: https://doi.org/10.7554/eLife.29173.015

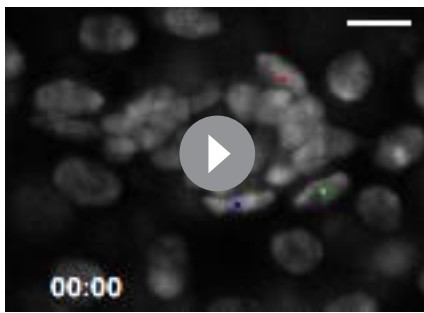

**Video 3.** Induction of epithelial cells into nBCs (2). The same developing secondary neuromast from *Video 2* is followed after a 10 min pause to exchange the medium. Epithelial cells (red, blue and green dots) in contact with the neural stem cell precursors start undergoing a significant change in their nuclear morphology and adopt a more elongated nuclear shape. These epithelial cells are transitioning into nBCs. Epithelial cells that divide during this process (blue dot) produce two cells with an elongated nuclear shape. The length of the movie is 24 hr. Stacks were taken every 20 min. Scale bar = 10 μm.
DOI: https://doi.org/10.7554/eLife.29173.016

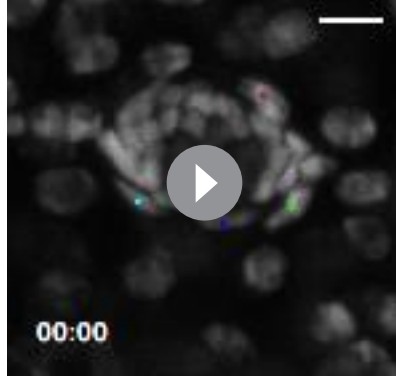

**Video 4.** Induction of epithelial cells into nBCs (3). The same developing secondary neuromast from *Video 3* is followed after a 10 min pause to exchange the medium. nBCs form the outer lineage of the neuromast organ (red, light and dark blue and green dots) surround the differentiating neural lineage cells. The characteristic elongated nuclear shape of nBCs is a stable feature and the transformation event seems to be complete. The length of the movie is 6 hr and 40 min. Stacks were taken every 20 min. Scale bar = 10 μm.
DOI: https://doi.org/10.7554/eLife.29173.017

observed that the neural lineage is maintained and the integrity of the organ is re-established when new border cells re-appear (*Figure 10B''*), a process that happens within a week after nBC removal (N = 8 neuromasts in four embryos). The presence of *K15* +nBCs in the regenerated neuromasts (*Figure 10B''*) suggests that these originate from a re-recruitment event from the skin epithelium, but we cannot exclude that they also originate from highly proliferative nBCs that were unlabelled and hence survived the 2-photon laser ablation. Taken together, these results demonstrate that nBCs are crucial for maintaining neuromast integrity and architecture and suggest they act as a physical niche for $K15^+$ neural stem cells.

## The neural lineage is necessary and sufficient for inducing nBCs

Neuromast organogenesis requires that cells deposited by the primordium (neuromast neural lineage) and skin epithelial cells come into contact. This could happen either by an active migration of cells deposited by the primordium towards pre-defined hot spots for epithelial induction or alternatively, by the local conversion of epithelial cells by the unspecified arrival of the neuromast neural lineage. Our previous observations suggest a leading role for the neural lineage, since left and right pLLs in medaka larvae often have an unequal number of neuromasts located in undetermined positions (*Seleit et al., 2017*). Irrespective of how many neural clusters are generated by the primordium, and regardless of the final position along the anterior-posterior axis, all mature neuromasts contain nBCs.

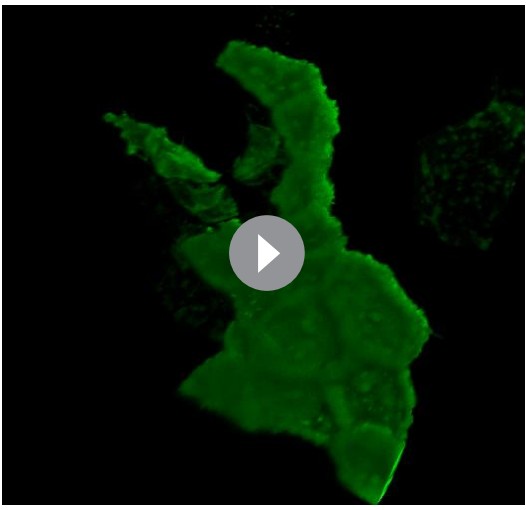

**Video 5.** 3D reconstruction of a membrane labelled nBC clone. A 3D projection of a mosaic larvae injected with (*K15*:mYFP). A clone of nBC is continuous with the suprabasal epithelial cell layer (N = 8 clones in five larvae).
DOI: https://doi.org/10.7554/eLife.29173.022

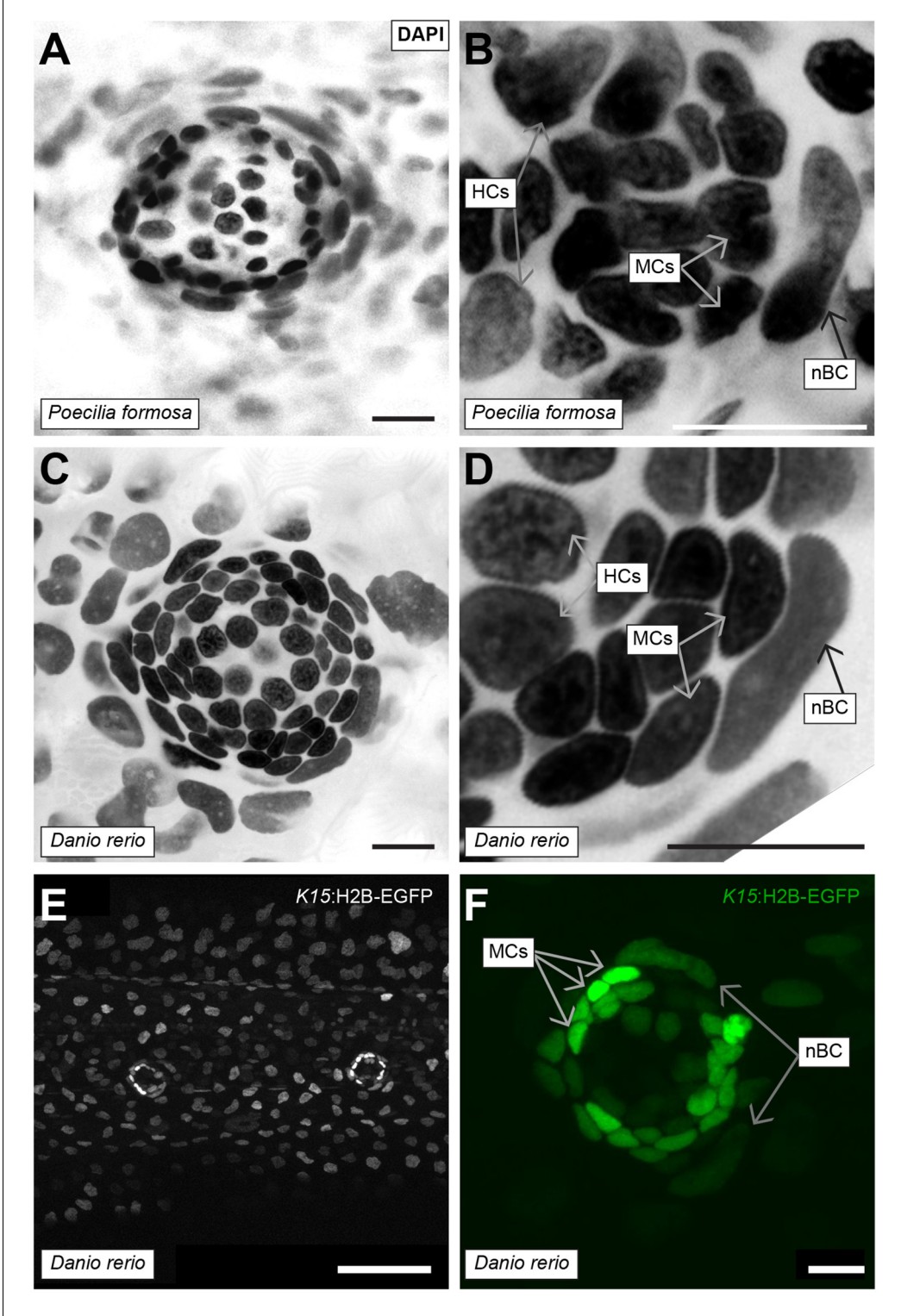

**Figure 9.** Conservation of nBCs in neuromasts of other teleost fish. (**A**) DAPI-stained neuromast *Peocilia formosa*, a close relative of medaka, reveals the presence of nBCs as assessed by their distinct morphology and relative location (N = 10 neuromasts in a single 1 month old fish). (**B**) Close-up on a *Peocilia formosa* neuromast reveals that nBCs are immediately adjacent to mantle cells (N = 10 border cells in four neuromasts). (**C**) DAPI-stained neuromast of *Danio rerio*, a distant relative of medaka, reveals the presence of nBCs as assessed by their distinct morphology and location (N=>5 neuromasts in three larvae). (**D**) Close-up of a zebrafish neuromast reveals that

*Figure 9 continued on next page*

*Figure 9 continued*
nBCs intimately associate with mantle cells (N=>5 neuromasts in three larvae). (E) Tg(*ol.K15*::H2B-eGFP) shows labelling of the skin epithelium and mantle cells in a 5 dph zebrafish larvae. (F) Close-up of a Tg(*ol.K15*::H2B-eGFP) zebrafish neuromast shows labelling of cells within the neural lineage and nBCs (N= 5 neuromasts in three larvae). Scalebars = 10 μm, except in panel E, Scalebar = 100 μm.
DOI: https://doi.org/10.7554/eLife.29173.024

We experimentally tackled the hierarchical organisation of the interacting tissues by two complementary approaches that result in fish with either ectopic or absent neuromasts along the pLL. The loss-of-function of *neurogenin-1* or *erbb* has been shown to result in the formation of ectopic neuromasts in zebrafish by neural lineage interneuromast cells that re-enter the cell cycle (*Grant et al., 2005*; *López-Schier and Hudspeth, 2005*; *Lush and Piotrowski, 2014*). The injection of Cas9 mRNA and two gRNAs directed against medaka *neurogenin-1* into Tg(*Eya1*:EGFP) or Tg(*Eya1*:H2B-EGFP) resulted in juveniles with an increased amount of pLL neuromasts (*Figure 11A* and *Figure 11—figure supplement 1*). We focused on the severest phenotypes, which contained twice the amount of neuromasts as compared to their control siblings (N = 58 pLL neuromasts in two larvae). Our results show that ectopic midline neuromasts are composed of neural lineage and nBCs (*Figure 11C–I*) (N = 33 neuromasts in two larvae). This suggests that the presence of the neural lineage is sufficient to drive ectopic induction of epithelial cells into border cells. Complementarily, ablation of the primordium before deposition of the last neuromast results in fish lacking the CNC (*Figure 11J,K*). The stereotypic position of the CNC constitutes an ideal set up to explore whether conversion of epithelial cells into border cells occurs even in the absence of the neural lineage. A detailed analysis of DAPI stained caudal fins revealed the absence of induced border cells on the ablated side, as opposed to their presence on the non-ablated, control side (*Figure 11J',K'*) (N = 4 CNCs of 4 fish). Altogether, our results demonstrate that the neural lineage is both necessary and sufficient to induce the conversion of epithelial cells into neuromast border cells in situ.

## Discussion

Here we identify a new type of neural stem cell (*Keratin15*[+]) that fulfils all criteria of stemness: it generates more neural stem cells, all neuromast neural progenitors and neurons during homeostasis and regeneration and additionally functions as the organ founder cell during post-embryonic organogenesis. We tracked these neural stem cells back to organogenesis to show that their precursors modify their environment to induce and recruit a new cell type to the forming organ.

### The origin of lineage commitment

Lineage commitment in stem cells could result from alternative scenarios. On the one hand, stem cells with different potencies can derive from a common pool of parental cells that acquire fate-restriction as development proceeds. This is the case for luminal (*K8*[+]) and myoepithelial (*K14*[+]) fate-restricted stem cells in the mammary gland that derive from a common pool of (*K14*[+]) embryonic multipotent cells (*Van Keymeulen et al., 2011*), and neural retina and pigmented epithelium (*Rx2*[+]) fate-restricted stem cells in the post-embryonic fish retina that derive from common embryonic retinal

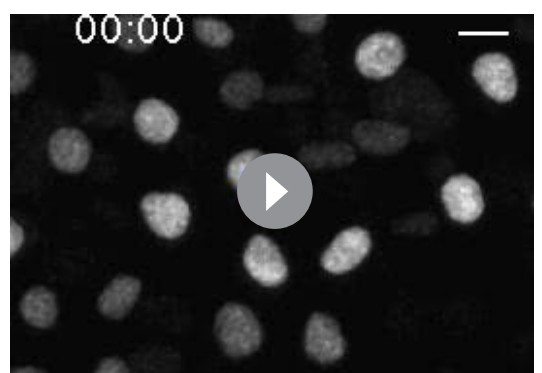

**Video 6.** Induction of epithelial cells into nBCs in zebrafish. A double Tg(ol.*K15*::H2B-eGFP)(*cxcr4b*::Cxcr4b-GFP) 24hpf embryo shows label in the skin epithelium (nuclear GFP) and the migrating primordium (membrane GFP). As the primordium deposits cells from its trailing end they dynamically associate with the overlying skin epithelium. A transformation of nuclear shape then occurs to produce a nBC cell (blue dot). Soon after deposition presumptive mantle cells start expressing *K15* and the neuromast progressively loses its *cxcr4b* fluorescent signal (N = 4 neuromasts in two embryos). Time in hours. Scale bar = 10 μm.
DOI: https://doi.org/10.7554/eLife.29173.023

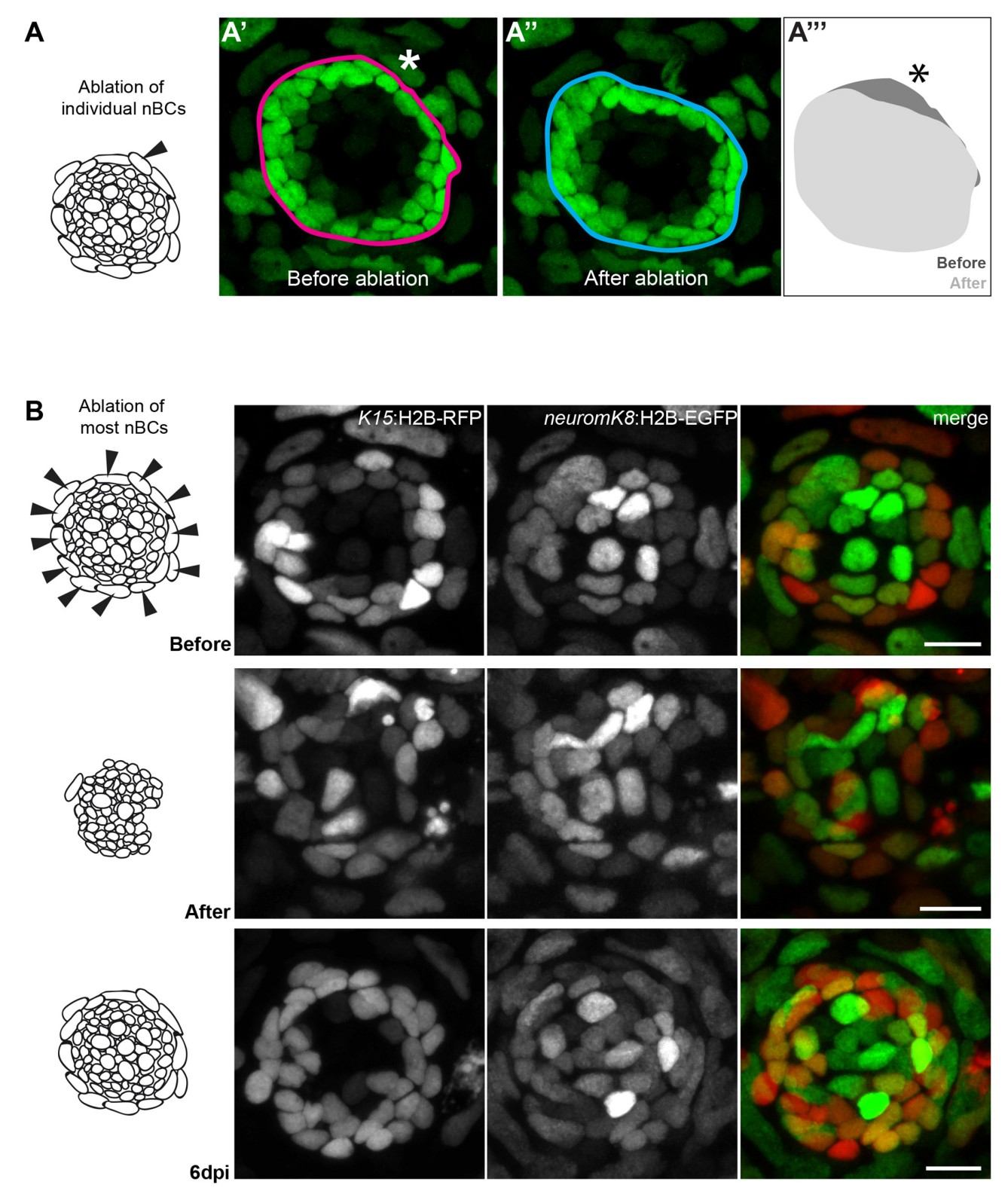

**Figure 10.** Ablation of nBCs disrupts organ architecture. (A–A''') Ablation of single nBCs in mature neuromasts (stage 39) of Tg(K15:H2B-EGFP) results in a local disruption of the organ architecture near the injury site. Mantle cells retract inwards towards the center of the neuromas resulting in a shape change of the organ (A'') (N = 10 neuromasts in seven larvae.) (B) Ablation of most nBCs in an immature neuromast of a stage 36 double Tg(K15::H2B-

*Figure 10 continued on next page*

*Figure 10 continued*

RFP) (*neurom^{K8}*:H2B-EGFP) results in a severe disruption of organ architecture. Iterative imaging reveals that new nBCs appear by 6 days post injury and organ architecture is re-established (N = 8 neuromasts in four fish). Scalebars = 10 µm.

DOI: https://doi.org/10.7554/eLife.29173.026

progenitors cells (*Centanin et al., 2014*; *2011*; *Reinhardt et al., 2015*). On the other hand, stem cells with independent embryonic origins could be brought together during early organogenesis to maintain different tissues in the same organ, as typically observed in the mammalian skin (*Fuchs et al., 2004*; *Ouspenskaia et al., 2016*) and more generally, in most composite organs. We have presented evidence that indicates a dual embryonic origin for medaka neuromasts. A neuromast neural lineage consists of mantle, support and hair cells surrounded by a second, independent lineage of neuromast border cells. Our findings position neuromasts as an approachable and minimalist system to address the coordination of independent lineages during homeostatic replacement, regeneration and post-embryonic organogenesis.

## Mantle cells during regeneration

We observed that a few remaining $K15^+$ mantle cells can reconstitute the entire neural lineage upon severe injury. Interestingly, the dynamics of neuromast regeneration seems to resemble the formation of secondary neuromasts during embryonic development (*Seleit et al., 2017*). We observe that upon injury the initial reaction of the surviving mantle cells is to coalesce which is followed by an increase in cell numbers. Only afterwards does differentiation take place and all three cell types of the neural lineage are reconstituted. This highly ordered transition suggests that before differentiation developing neuromasts must attain a critical number of cells, as we observed during embryonic development (*Seleit et al., 2017*). Intriguingly, the efficient regeneration of neuromasts in medaka contrasts with its reported inability to adequately regenerate its heart upon mechanical injury (*Ito et al., 2014*). It has recently been shown that regeneration enhancer elements exist in zebrafish and can drive tissue specific regenerative responses to injury (*Kang et al., 2016*).

A differential selective pressure could impact those TREEs (tissue-specific regeneration enhancer elements) in a species-specific manner, and can therefore have divergent effects on the regenerative capacity of specific organs. A comparative genomic approach on TREEs among teleosts could contribute to the understanding of their differential regenerative capacities, and our results provide yet another module to study the evolution of regeneration responses across fish.

## Mantle cells as neuromast stem cells

Neuromasts in fish can replace lost HCs during their entire life, which has been taken as an indication that neuromasts should contain neural stem cells. Numerous short-term studies in zebrafish, however, have reported that both during homeostasis and regeneration HCs are produced by post-mitotic hair cell precursors (*Hernández et al., 2007*; *Kniss et al., 2016*) and proliferation of support cells (*Hernández et al., 2007*; *López-Schier and Hudspeth, 2006*; *Pinto-Teixeira et al., 2015*; *Romero-Carvajal et al., 2015*). In the past years, several studies have reported the existence of compartments or territories within the neuromasts that permit either self-renewal of support cells or

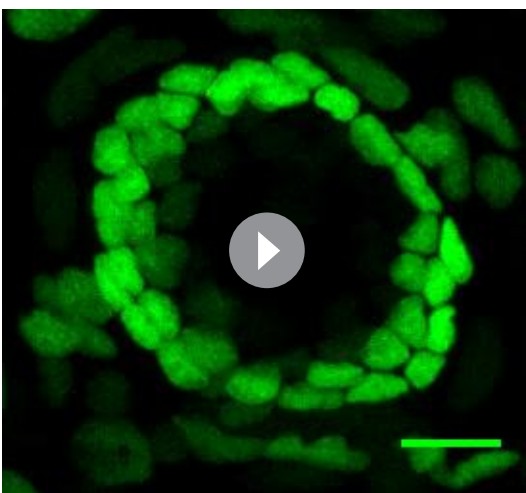

**Video 7.** Local architectural changes upon ablation of a single nBC. Before and after the ablation of an individual nBC in a Tg(*K15*:H2B-EGFP) neuromast. Mantle cells directly adjacent to the ablated nBC retract inwards. The organ architecture is locally disrupted producing an organ with a notch-shape (N = 10 neuromasts in seven larvae.) Scale bar = 10 µm.

DOI: https://doi.org/10.7554/eLife.29173.025

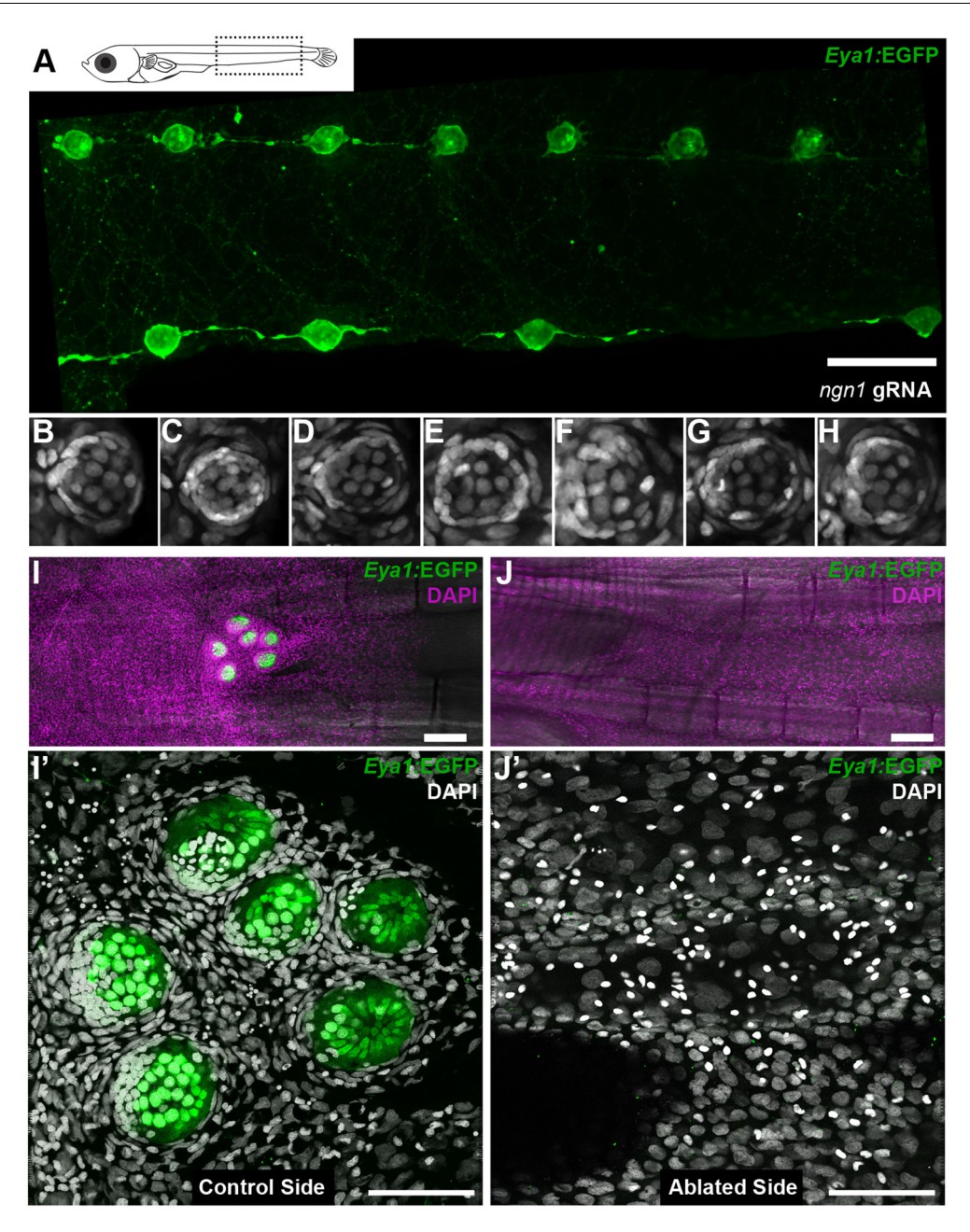

**Figure 11.** Arrival of neural neuromast cells is necessary and sufficient to convert epithelial cells into nBCs. (**A**) Anti-GFP staining of a Tg(*Eya1*:EGFP) embryo at stage 42 that was injected with *CAS9* mRNA and *ngn1* gRNAs reveals the formation of ectopic pLL midline neuromasts (N = 58 pLL neuromasts in two larvae). (**B–H**) DAPI images of seven consecutive midline neuromasts show that all of them contain nBCs. (**I–J'**) Absence of neural lineage prevents the induction of the skin epithelium. While each neuromast of a control CNC (**I**) contains nBCs (**I'**), early ablation of neural lineage in neuromast[P0] results in no CNC (**J**) and no skin epithelium induction (**J'**) (n = 4 CNCs of 4 fish). Wide scalebars are 100 µm (**A, I**) and thinner scalebars are 50 µm (**I', J'**).

DOI: https://doi.org/10.7554/eLife.29173.027

The following figure supplement is available for figure 11:

**Figure supplement 1.** Increased number of neuromasts in ngn1 gRNA injected medaka.
DOI: https://doi.org/10.7554/eLife.29173.028

differentiation into HCs (*Pinto-Teixeira et al., 2015*; *Romero-Carvajal et al., 2015*; *Wibowo et al., 2011*). Despite the detailed knowledge on short-term aspects of regeneration and homeostatic replacement, the existence and identity of long-term neuromast stem cells has remained elusive. Using genetic tools to irreversibly label cells and their progeny, we followed long-term lineages to show the existence of neural stem cells in the organ and uncover their identity. Using ubiquitous drivers for recombination (either *Ubiquitin*:Cre$^{ERT2}$ or *Hsp70*:$_{nls}$Cre) we observed that all long-term clones in the neural lineage contain mantle cells, suggesting that these are necessary for the clone to be maintained. Additionally, a cell type-specific driver for mantle cells (*K15*:Cre$^{ERT2}$) proved these cells to function as neural stem cells under homeostatic conditions. Our results indicate that *K15$^+$* mantle cells in medaka work as *bona fide* neural stem cells, maintaining clones that expand within neuromasts and generating new organs during post-embryonic life.

## Induced fate conversion

We report that upon arrival of neuromast neural lineage cells deposited by the migrating primordium (the case of ventral neuromasts) or originated by coalescence of inter-neuromast cells (the case of midline neuromasts), epithelial cells follow a morphological and molecular transition that results in the generation of border cells. This event seems to be restricted to the initial phases of organ formation and involves a small number of cells. Our 4D analysis of the epithelial-to-border cell transition has revealed that border cells can divide after conversion to generate two border cells, which was confirmed by live imaging and short term lineage experiments in medaka. Labelled cells can generate large clones that involve all border cells in every neuromast of the CNC, indicating long-term stability of labelled clones. Altogether, we have shown that neural stem cell precursors trigger the induction of nBCs from suprabasal skin epithelial cells, which will in turn associate with them to form mature neuromast organs. The analysis of cell membranes during the conversion phase has revealed that skin epithelial cells change their morphology mirroring what has been observed for the induced nuclei. Notably during the induction phase, the neural stem cell precursors extend numerous cytoplasmic protrusions that dissapear once the stem cell — niche interaction has been established in mature organs. Cytoplasmic extensions have been reported in other systems as a way in which the niche signals to stem cells such as nanotubes in *Drosophila* (*Inaba et al., 2015*), opening the possibility that these processes are indeed a communication system between the two lineages during neuromast formation. Further experiments disrupting these cytoplasmic protrusions will contribute to defining their role in organogenesis. Overall, we believe that our findings position neuromasts as a new paradigm to dynamically study the inductive behaviour that cell types exert on each other during organogenesis.

The formation of the lens in the vertebrate retina was the first reported case of tissue induction. There, the developing optic vesicle (composed of neural progenitors) contacts the surface ectoderm to induce a lens placode from previous ectoderm cells (*Cvekl and Ashery-Padan, 2014*; *Gunhaga, 2011*), (*Bailey et al., 2006*). The case that we report here shares the same rationale and interestingly, the cells involved express the same molecular markers. Neuromast neural lineage cells are *Eya1$^+$* as retinal progenitors, and they induce a cell fate change in *K15$^+$* ectodermal cells. The similar molecular identity of the cell types involved, in which *Eya1$^+$* cells operate as the inducer and *K15$^+$Eya1$^-$* cells as responders, invites to question whether similar molecular mechanisms are involved in the event. The identity and nature of the inducer cells — in both cases neural progenitors responsible for generating neurons that define the organ's function — indicate a shared hierarchical organisation during organogenesis of sensory systems. It would be interesting to address whether other sensory systems with a similar surface location and common molecular markers (e.g. the ampular system) follow the same rationale during organogenesis and ultimately, whether proximate interaction, or induction, constitutes an ancient signature for the establishment of sensory systems.

## nBCs as a niche for neural stem cells

The stem cell niche is key for stem cell maintenance in plants, invertebrates and vertebrates.. The concept of the stem cell niche was introduced by Schofield during the late 70's (*Schofield, 1978*), and since then is defined as the stem cell microenvironment and architecture (*Fuchs et al., 2004*; *Morrison and Spradling, 2008*; *Nystul and Spradling, 2006*; *Scadden, 2014*). The niche

influences the behaviour of stem cells by a variety of mechanisms including direct signalling (*Tulina and Matunis, 2001*; *Xie and Spradling, 1998*), in a way that stem cells that keep a close connection to the niche are destined to maintain their stemness while their displaced progeny start to differentiate. The complexity of the stem cell niche varies depending on stem cell type, spanning from the relatively simple case of *Drosophila* gonads to the extraordinarily vast hemato-poietic niche. In the fly testis, germ stem cells are in contact with Hub cells (niche cells), and this interaction defines stemness through a series of well defined signalling pathways (*Tulina and Matunis, 2001*; *Xie and Spradling, 1998*). In contrast, many cell types in the bone marrow have been reported as hematopoetic stem cell (HSC) regulators and contributors to the hematopoietic niche, including endothelial cells, LepR+, adipocytes, Schwann cells, nerve fibers and even macro-phages and megakaryocytes (*Bruns et al., 2014*; *Christopher et al., 2011*; *Méndez-Ferrer et al., 2008*; *Scadden, 2014*; *Yamazaki et al., 2011*; *Zhou et al., 2017*). Many examples have been reported on how stem cells can act upon changes in their microenvironment. Mouse embryonic stem cells will self-renew upon culture conditions, generate any differentiated cell type if trans-planted into a blastula but produce teratomas if implanted in an adult (reviewed in (*Fuchs et al., 2004*). Luminal stem cells in the mammary gland are bipotent in an embryonic niche, unipotent in the adult niche, and bipotent again upon loss of their basal counterpart (*Van Keymeulen et al., 2011*). Neoblasts in planarians generate the cell types of their surroundings, but upon excision of a major part of the body they can produce distal cell types and even generate an entire new organism (*Gurley et al., 2008*; *Sánchez Alvarado, 2003*; *Wagner et al., 2011*). The surrounding microenvironment of a stem cell therefore defines crucial aspects of its identity.

The physical proximity and intimate contacts of neuromast border cells and mantle cells position nBCs as a strong candidate to constitute the neural stem cell niche of the organ. Indeed, we observe in medaka that MCs maintaining contact with nBCs continue expression of the neural stem cell marker ($K15^+$) and display stem cell behaviour, that is generation of long-term clones. Additionally, the removal of individual niche cells has local effects on the stem cells that were attached to them, while major ablations affect the overall architecture of the organ. In short, nBCs are part of the life-long direct microenvironmet of mantle stem cells and their relative location fulfills the requirements expected for niche cells – cells in contact remain stem cells, daughters that are located tangentially also become stem cells, and daughter cells moving radially inwards start differentiation programmes to eventually generate neurons. We believe that the simplicity of neuromast organisation, the abun-dance of organs per experimental animal and the possibility of combining physical manipulations and live-imaging position the neuromast as a highly complementary model to explore stem cell–niche interactions.

Further molecular characterisation of nBCs is needed to identify cell type-specific drivers, which in turn will facilitate analysing their impact on neural stem cells on the long-term scale. Additional abla-tions, for instance removing all mantle cells, are required to address whether nBCs could confer *K15* expression and long-term stem cell behavior to other cells types in the organ, as it was demon-strated for other systems (*Tetteh et al., 2016*). The conservation of the same organ architecture and cell types in zebrafish will allow following dynamic aspects of niche—stem cell interaction in a model better suited for continuous 4D approaches.

Most niche cells have a different lineage than their respective stem cells (*Fuchs et al., 2004*), which raises fundamental developmental questions such as how do these cell types come together (*Tamplin et al., 2015*), whether they are formed simultaneously or sequentially (*Ouspenskaia et al., 2016*) and if there is a hierarchy organising their interaction. Our results suggest that during develop-ment, the neural lineage induces the formation of its own niche by fate conversion of neighbouring epithelial cells. The induction of a transient, short-lived niche by HSCs in zebrafish was recently reported to occur by major morphological remodelling of perivascular endothelial cells (*Tamplin et al., 2015*). We report a similar morphological transition during sensory organ formation that notably results in a life-long, permanent niche. This model in which niche formation is triggered by arriving neural precursors seems appropriate for a system in which the number and location of organs is plastic and not genetically defined. Whether the same rationale applies to niche formation upon the

arrival of migrating malignant cells in pathological cases is a clinically relevant question that remains to be elucidated.

**Key resources table**

| Reagent type (species) or resource | Designation | Source or reference | Identifiers | Additional information |
|---|---|---|---|---|
| strain (*Oryzias latipes*) | Cab | other | | medaka Southern wild type population |
| strain (*O. latipes*) | Tg(*Eya1*:H2B-EGFP) | PMID: 28087632 | | |
| strain (*O. latipes*) | Tg(*Eya1*:EGFP) | PMID: 28087632 | | |
| strain (*O. latipes*) | GaudíUbiq.iCre | PMID:25142461 | | |
| strain (*O. latipes*) | GaudíHsp70.A | PMID:25142461 | | |
| strain (*O. latipes*) | GaudíRSG | PMID:25142461 | | |
| strain (*O. latipes*) | Tg(*K15*:mYFP) | this paper | | *K15* sequence by PCR using fosmid GOLWFno691_n05 (NBRP Medaka) as template |
| strain (*O. latipes*) | Tg(*K15*:H2B-EGFP) | this paper | | *K15*sequence by PCR using fosmid GOLWFno691_n05 (NBRP Medaka) as template |
| strain (*O. latipes*) | Tg(*K15*:H2B-RFP) | this paper | | *K15* sequence by PCR using fosmid GOLWFno691_n05 (NBRP Medaka) as template |
| strain (*O. latipes*) | Tg(*K15*:ERT2Cre) | this paper | | *K15* sequence by PCR using fosmid GOLWFno691_n05 (NBRP Medaka) as template |
| strain (*O. latipes*) | Tg(*neuromK8*:H2B-EGFP) | this paper | | 0.5 Kb *Keratin8* sequence PMID: 18544450 |
| strain (*Danio rerio*) | AB | other | | Wildtype zebrafish strain |
| strain (*D. rerio*) | Tg(*o.l.K15*:H2B-EGFP) | this paper | | *K15*:H2B-EGFP plasmid from medaka (this paper) |
| strain (*D. rerio*) | Tg(*cxcr4b*:Cxcr4b-GFP) | PMID: 24067609 | | |
| antibody | a-EGFP (rabbit IgG polyclonal) | Invitrogen (now Thermo Fischer) | CAB4211; RRID: AB_10709851 | 1:750 |
| antibody | a-EGFP (chicken IgY polyclonal) | life technologies | A10262; RRID: AB_2534023 | 1:750 |
| antibody | a-Sox2 (rabbit polyclonal) | GeneTex | GTX101506 | 1:100 |
| antibody | a-DsRed (rabbit polyclonal) | ClonTech | 632496 | 1:500 |
| antibody | Alexa 488 goat a-Rabbit | Invitrogen (now Thermo Fischer) | A-11034 | 1:500 |
| antibody | DyLight549 Goat anti Rabbit IgG ML | Jackson | 112-505-144 | 1:500 |
| antibody | Alexa 488 donkey a-chicken | Invitrogen (now Thermo Fischer) | 703-545-155 | 1:500 |
| antibody | Alexa 647 goat a-Rabbit | Life Technologies | A-21245 | 1:500 |
| recombinant DNA reagent | *K15*:H2B-EGFP (plasmid) | this paper | | Vector with I-SceI meganuclease sites |
| recombinant DNA reagent | *K15*:mYFP (plasmid) | this paper | | Vector with I-SceI meganuclease sites |
| recombinant DNA reagent | *K15*:H2B-RFP (plasmid) | this paper | | Vector with I-SceI meganuclease sites |
| recombinant DNA reagent | *K8*:H2B-EGFP) (plasmid) | this paper | | Vector with I-SceI meganuclease sites |

*Continued on next page*

*Continued*

#### Key resources table

| Reagent type (species) or resource | Designation | Source or reference | Identifiers | Additional information |
|---|---|---|---|---|
| recombinant DNA reagent | K15:CreERT2 (plasmid) | this paper | | Vector with I-SceI meganuclease sites |
| recombinant DNA reagent | K15:PRLexA OPLex:nlsCRE (plasmid) | this paper | | Vector with I-SceI meganuclease sites |
| recombinant DNA reagent | DNA oligo 1 for ngn-1 | this paper | | gRNAngn1-1F: TAGGTTCTCAGTGCTCGAGTCCGG; gRNAngn1-1R: AAACCCGGACTCG AGCACTGAGAA |
| recombinant DNA reagent | DNA oligo 2 for ngn-1 | this paper | | sgRNAngn1-2F: TAGGTCTGCGATG CGGATGGTCT; sgRNAngn1-2R: AA ACAGACCATCCGCATCGCAGA |
| sequence-based reagent | CAS9 mRNA | home-made | | |
| sequence-based reagent | gRNA 1 for ngn-1 | this paper | | UUCUCAGUGCUCGAGUCCGGCGG |
| sequence-based reagent | gRNA 2 for ngn-1 | this paper | | UUCUCAGUGCUCGAGUCCGGCGG |
| sequence-based reagent | PCR primer for K15 | this paper | | fwd: ACTGACTCGAGACCAAAG GAAAGCAGATGAA; rev: ACTG ACTCGAGTTGTGCAGTGTGGTC GGAGA |
| chemical compound, drug | tamoxifen | Sigma | T5648 | |
| chemical compound, drug | tricaine | Sigma-Aldrich | A5040-25G | |
| software, algorithm | CC-Top | PMID:25909470 | | |
| software, algorithm | Ensemble | Public | | |
| other | DAPI | Roth | | final concentration of 5 ug/l |

# Materials and methods

## Fish stocks and generation of transgenic lines

Medaka (*Oryzias latipes*) and zebrafish (*Danio rerio*) stocks were maintained as closed stocks in a fish facility built according to the local animal welfare standards (Tierschutzgesetz §11, Abs. 1, Nr. 1), and animal experiments were performed in accordance with European Union animal welfare guidelines. The facility is under the supervision of the local representative of the animal welfare agency. Fish were maintained in a constant recirculating system at 28°C with a 14 hr light/10 hr dark cycle (Tierschutzgesetz 111, Abs. 1, Nr. 1, Haltungserlaubnis AZ35–9185.64 and AZ35–9185.64/BH KIT).

The strains and transgenic lines used in this study are: Cab (medaka Southern wild type population), Tg(*Eya1*:EGFP), Tg(*Eya1*:H2B-EGFP) (*Seleit et al., 2017*), Gaudi<sup>Ubiq.iCre</sup>, Gaudi<sup>Hsp70.A</sup>, Gaudi<sup>RSG</sup> (*Centanin et al., 2014*), Tg(cxcr4b:Cxcr4b-EGFP (*Donà et al., 2013*). The following transgenic lines were generated for this manuscript by I-SceI mediated insertion, as previously described (*Rembold et al., 2006*): Tg(*K15*:mYFP), Tg(*K15*:H2B-EGFP), Tg(*K15*:H2B-RFP), Tg(*K15*:Cre<sup>ERT2</sup>), Tg (*neurom*<sup>K8</sup>:H2B-EGFP) in medaka, and Tg(ol*K15*:H2B-EGFP) in zebrafish.

## Generation of the constructs *K15*:mYFP, *K15*:H2B-EGFP, *K15*:H2B-RFP

A 2.3 kb fragment upstream of the medaka *Keratin15* ATG was amplified by PCR using specific primers flanked by XhoI sites (forward: ACTGACTCGAGACCAAAGGAAAGCAGATGAA; reverse: AC TGACTCGAGTTGTGCAGTGTGGTCGGAGA) using the fosmid GOLWFno691_n05 (NBRP Medaka) as template. The PCR fragment was cloned into an I-SceI vector already containing mYFP, and subcloned from there into I-SceI vectors containing either H2B-EGFP or H2B-RFP.

## Generation of the construct $K15$:Cre$^{ERT2}$

The 2.3 Kb *Keratin15* promoter was cut with XhoI from the *K15*:mYFP plasmid and cloned upstream of Cre$^{ERT2}$, replacing the ubiquitin promotor in *Ubiquitin*:Cre$^{ERT2}$ (*Centanin et al., 2014*).

## Generation of the construct $K8$:H2B-EGFP

The 0.5 Kb *Keratin8* promoter from zebrafish (*Emelyanov and Parinov, 2008*) was sub-cloned into an I-SceI vector containing H2B-EGFP via KpnI/AscI. Among the founders obtained, one expressed high levels of H2B-EGFP in both skin epithelium and neuromasts and was used to establish Tg(*neurom$^{K8}$*:H2B-EGFP). Other founders from the same injection did not share the expression in the neuromasts.

## Generation of clones

Fish from the Gaudí$^{RSG}$ transgenic line were crossed with either Gaudí$^{Ubiq.iCre}$ or Tg(K15:Cre$^{ERT2}$). The progeny from these crosses was induced with a 5 μM tamoxifen (T5648 Sigma-Aldrich Chemie GmbH, Germany) solution for 3–12 hr and afterwards washed extensively with ERM. When Gaudí$^{RSG}$ fish were crossed to Gaudí$^{Hsp70.A}$, double transgenic embryos were heat-shocked using ERM at 42°C and transferred to 37°C for 1–3 hr. Clones generated by injection of DNA into the 1–2 cell stage were prepared as previously stated (*Rembold et al., 2006*).

## Antibodies and staining

Immunofluorescence staining were performed as previously described (*Centanin et al., 2014*). The primary antibodies used in this study were rabbit a-GFP, chicken a-GFP (Invitrogen, by thermo Fisher Scientific, Lithuania, both 1/750), rabbit a-DsRed (Clontech, TBUSA, Mountain View, CA94043, 1/500) and rabbit a-Sox2 (GeneTex, Inc., Irvine, CA92606, 1/100). Secondary antibodies were Alexa 488 a-Rabbit, Alexa 546 a-Rabbit, Alexa 647 a-Rabbit (Invitrogen, all 1/500). DAPI was used in a final concentration of 5 ug/l.

## Imaging and image analysis

### Preparation of samples for live imaging

Embryos were prepared for live-imaging as previously described (*Seleit et al., 2017*). Briefly, we used a 20 mg/ml as a 20x stock solution of tricaine (Sigma-Aldrich, A5040-25G) to anaesthetise dechorionated embryos and mounted them in low melting agarose (0,6 to 1%). Imaging was done on glass-bottomed dishes (MatTek Corporation, Ashland, MA 01721, USA).

### Preparation of fixed samples

Stained samples were mounted in glycerol 80% on glass slides. Samples that required imaging from both sides of the fish were mounted between two cover slides using a minimal spacer.

### Imaging

Anaesthetised embryos were screened using an Olympus MVX10 binocular coupled to a Leica DFC500 camera. For the acquisition of high quality images, we used a Nikon AZ100 scope coupled to a Nikon C1 confocal, or the laser-scanning confocal microscopes Leica TCS SPE and Leica TCS SP5 II. When imaging living samples over long-term time lapse, a Microscope Slide Temperature Controller (Biotronix GmbH, Germany) was used and the temperature was fixed to 28°C. Time-lapse imaging of epithelial cells was done over a period of 72 hr using an EMBL MuVi-SPIM (*Krzic et al., 2012*) with two illumination objectives (10x Nikon Plan Fluorite Objective, 0.30 NA) and two detection objectives (16X Nikon CFI LWD Plan Fluorite Objective, 0.80 NA). Embryos were placed in glass capillaries using 0,6% low melting agarose. Imaging was performed at room temperature. All subsequent image analysis was performed using standard Fiji software.

### Image analysis

We used the free standard Fiji software for analysis and editing of most images. Stitching was performed automatically using 2D and 3D stitching plug-ins on ImageJ or using Adobe Photoshop to align images manually.

## Clonal lineage analysis

For short-term lineage experiments, clones were defined as 1–4 adjacent cells and up to three clones per organ were considered (*Figure 3—figure supplement 2*). Cell type annotation was done based on nuclear morphology and position within the neuromast. For each imaging time point the cell type composition of clones was annotated in some cases making use of the '3 D viewer' tool in the Leica SpE software.

In long-term lineage analysis, we quantified clones bigger than 4 cells and divided labelled neuromasts into labelled neural lineage, nBCs or co-labelled. As stated in the text we considered CNCs in which less than 75% of neuromasts per cluster were labelled for our quantifications of occurance of labelling in the separate lineages.

## EM

10dpf Tg(*Eya1*:EGFP) embryos were fixed in 2.5% glutaraldehyde and 4% paraformaldehyde in 0.1M PHEM buffer for 30 min at room temperature and at 4 °C overnight. After rinsing in buffer, embryos were imaged under a Leica MZ 10F stereo microscope for fluorescent imaging (Leica Microsystems, Vienna) to localise neuromasts. The samples were further fixed in 1% osmium in 0.1M PHEM buffer, washed in water, and incubated in 1% uranylacetate in water overnight. Dehydration was done in 10 min steps in an acetone gradient followed by stepwise Spurr resin infiltration at room temperature and polymerization at 60 °C. The resulting blocks were trimmed around the neuromast to get longitudinal sections of the nBCs surrounding the mantle cells and sectioned using a leica UC6 ultramicrotome (Leica Microsystems Vienna) in 70 nm thin sections. The sections were placed on formvar coated slot grids, post-stained and imaged on a JEOL JEM-1400 electron microscope (JEOL, Tokyo) operating at 80 kV and equipped with a 4K TemCam F416 (Tietz Video and Image Processing Systems GmBH, Gautig).

## 2-Photon laser ablations

### Ablation of neuromast[P0]

We used a multi-photon laser coupled to a Leica TCS SP5 II microscope to perform specific ablations of the neuromast[P0] on Tg(*Eya1*:EGFP) embryos. We chose the option 'Area ablations' and used the 880 nm wavelength with a laser power ranging from 25% to 30%. The absence of the neuromast[P0] was checked immediately after the ablation and confirmed 24 hr later.

### Ablation of neural lineage in neuromasts

We used a TriM Scope 2-photon microscope (LaVision BioTec, Germany) as previously described (*Seleit et al., 2017*). The 2-photon module was mounted on a Nikon FN-1 upright microscope combined with a Chameleon Ultra II femtosecond Ti:Sa laser (Coherent, Dieburg, Germany). Ablations were preformed on late Tg(*K15*:H2B-RFP) Tg(*Eya1*:EGFP) and Tg(*K15*:H2B-RFP) Tg(*neuromK8*:H2B-EGFP) double transgenic embryos, using a 740 nm wavelength. Laser power ranged from 150 to 700 mW and was adjusted depending on the position of target neuromasts and the scope of the injury.

### Ablation of nBCs in neuromasts

We used a multi-photon laser coupled to a Leica TCS SP5 II microscope to perform nBC ablations in Tg(*k15*:eGFP) and double transgenic Tg(*K15*:H2BRFP) Tg(*neuromK8*:H2B-GFP) embryos. We chose the option 'point ablations' and used 880 nm wavelength with a laser power ranging from 25–30% for 250–600 ms. The targeted cells were immediately checked for signs of burst nuclei and for the experiment with double transgenics imaged iteratively post-injury.

## Mosaic loss-of-function of ngn1

We used the Ultracontig 115 (NBRP Medaka) to design gRNAs targeting the coding sequence of medaka *ngn1*. Two gRNAs were selected (ngn1-1: UUCUCAGUGCUCGAGUCCGGCGG, and ngn1-2: UUCUCAGUGCUCGAGUCCGGCGG) using the freely available CCTop (*Stemmer et al., 2015*). gRNA synthesis was done as previously reported (*Stemmer et al., 2015*) using the following oligos:

gRNAngn1-1F: TAGGTTCTCAGTGCTCGAGTCCGG
gRNAngn1-1R: AAACCCGGACTCGAGCACTGAGAA
sgRNAngn1-2F: TAGGTCTGCGATGCGGATGGTCT

sgRNAngn1-2R: AAACAGACCATCCGCATCGCAGA

Tg(*Eya1*:EGFP) or Tg(*Eya1*:H2B-EGFP) embryos were injected at the 1 cell stage with a solution containing 15 ng/µl of each gRNA and 150 ng/µl of CAS9 mRNA. The resulting embryos were selected for the presence of ectopic neuromasts at late embryonic stages.

## Quantifying morphological changes

Circularity was used to measure the transition of shape observed between epithelial cells and nBCs. Circularity = $4\pi(area/perimeter^2)$, a circularity value of 1.0 denotes a perfect circle. As the shape deforms away from a circle its circularity value decreases. Standard Fiji software was used to calculate the circularity of 20 nuclei and cellular membranes of epithelial cells and 20 nuclei and cellular membranes of nBCs obtained from mosaic clones of fish co-injected with the *K15*:H2BRFP and *K15*:mYFP plasmids. The generation of boxplots was done using standard R software.

## Acknowledgements

We thank T Piotrowski, S Lemke, J Wittbrodt and M Allende for scientific inputs on earlier versions of this manuscript, and K Lust, A Gutierrez-Triana, T Oskarsson and Centanin lab members for active discussions on the project. We are grateful to M Schartl for providing fixed samples of *Poecilia formosa*, D Gilmour for sharing Tg(*cxcr4b*:Cxcr4b-GFP), J Mateo Cerdan for help on the identification of the *Keratin 15* promoter, NBRP Medaka for sharing the fosmid containing *Keratin15*, U Engel for support with microscopes, and C Funaya, S Gold and S Hillmer from the EMCF facility at Heidelberg University for great technical assistance in the preparation and processing of electron microscopy data. We thank R Bump for the analysis of clones on the CNCs, and E Leist, A Sarraceno and M Majewski for assistance regarding fish maintenance. This work has been funded by the Deutsche Forschungsgemeinschaft (German Research Foundation, DFG) via the Collaborative Research Centre SFB873 (subproject A11 to LC) and the Cluster of Excellence Cellular Networks (Cell Networks) (to ND).

## Additional information

### Funding

| Funder | Grant reference number | Author |
|---|---|---|
| Deutsche Forschungsgemeinschaft | SFB 873 | Ali Seleit<br>Isabel Krämer<br>Elizabeth Mayela Ambrosio<br>Julian Stolper<br>Lazaro Centanin |

The funders had no role in study design, data collection and interpretation, or the decision to submit the work for publication.

### Author contributions

Ali Seleit, Conceptualization, Data curation, Formal analysis, Validation, Investigation, Visualization, Methodology, Writing—original draft, Writing—review and editing; Isabel Krämer, Data curation, Formal analysis, Validation, Investigation, Visualization, Writing—original draft, Writing—review and editing; Bea F Riebesehl, Data curation, Writing—review and editing; Elizabeth M Ambrosio, Data curation, Investigation, Visualization, Writing—review and editing; Julian S Stolper, Visualization, Writing—review and editing, Generation of transgenic lines; Colin Q Lischik, Mounting and data acquisition for light sheet microscopy; Nicolas Dross, Two-photon laser ablations (on TriM Scope 2-photon microscope); Lazaro Centanin, Conceptualization, Supervision, Funding acquisition, Investigation, Visualization, Methodology, Writing—original draft, Project administration, Writing—review and editing

### Author ORCIDs

Lazaro Centanin https://orcid.org/0000-0003-3889-4524

### Ethics

Animal experimentation: Medaka (Oryzias latipes) and zebrafish (Danio rerio) stocks were maintained as closed stocks in a fish facility built according to the local animal welfare standards (Tierschutzgesetz §11, Abs. 1, Nr. 1), and animal experiments were performed in accordance with European Union animal welfare guidelines. The facility is under the supervision of the local representative of the animal welfare agency. Fish were maintained in a constant recirculating system at 28°C with a 14 h light/10 h dark cycle (Tierschutzgesetz 111, Abs. 8 1, Nr. 1, Haltungserlaubnis AZ35-9185.64 and AZ35-9185.64/BH KIT).

### Decision letter and Author response

Decision letter https://doi.org/10.7554/eLife.29173.029
Author response https://doi.org/10.7554/eLife.29173.030

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
