## [Decision Letter]

Thank you for submitting your article "Neural stem cells induce the formation of their physical niche during organogenesis" for consideration by *eLife*. Your article has been reviewed by three peer reviewers, and the evaluation has been overseen by a Reviewing Editor (Tanya Whitfield) and Fiona Watt as the Senior Editor. The reviewers have opted to remain anonymous.

The reviewers have discussed the reviews with one another and the Reviewing Editor has drafted this decision to help you prepare a revised submission.

Seleit et al., have used short and long term lineage tracing to address the very interesting question of the origin of the stem cells that are necessary to replace: (i) hair cells during homeostasis and (ii) almost entire neuromasts during ablation and regeneration in the lateral line system of the Medaka. In addition, they have identified a new type of cells in the neuromasts, which they named neuromast Border cells (nBC) and which they show have a strictly different origin than the other neuromast cells. They propose that nBCs could serve as a niche for the neuromast stem cells.

All three reviewers commented on the merits of the study, including the long-term imaging, confirmation that mantle cells are the stem cells for the neuromast, and interesting observations concerning the border cells and their intimate contacts with the mantle cells. However, it was felt that some further work was necessary to strengthen the conclusions, and especially to support the interpretation that the border cells form a niche for the mantle (stem) cells of the neuromast. The essential suggested revisions are:

1) Reviewer 2 requests an additional video that includes a membrane label.

2) There was some discussion amongst the reviewers about experimental work that could be done to demonstrate a functional (niche) role for the border cells. As you will see, reviewer 1 comments on the current lack of experimental support for this interpretation, and ablation studies have been suggested by reviewer 2. It was acknowledged that these might be challenging to do, but it was felt that some attempt should be made to address this. It was also felt that unless further experimental support could be provided here, the authors should tone down their interpretations about the niche and alter the wording in the title.

3) In the Discussion, there could be more emphasis on what our understanding from other model systems has revealed about what constitutes a stem cell niche, why it is important and how the neuromast might serve as a useful model for the exploration of their formation and maintenance. It might also help discuss what has been learnt so far in this system and what additional observations might need to be made to firmly establish the neuromast as a model for self-organization of a stem cell niche.

4) Please give all details of quantification and statistical analysis, as requested by reviewer 2.

5) Please provide a brief explanation for the Gaudi methodology, as requested by two of the reviewers.

Reviewer #1:

This is a very nice study, the lineage tracing data are very clear and the light and electron microscopy data very informative. In addition, the paper is very clearly written and pleasant to read. Yet, I have two major concerns about the manuscript.

1) Evidence showing that nBCare an integral part of the neuromasts is missing.

The authors describe the newly identified "nBC" as an integral part of the neuromasts. Since those cells have a different origin (the overlying epidermis, the authors show that very clearly), this is not clear to me at which point the nBC become part of the neuromasts. That the nBCare changing shape in the plane of the epidermis as neuromasts get deposited and mature is clear, but to my point of view, this is not sufficient to claim that they are an integral part of the neuromasts. If this is not the case, the nBCs should be considered as cells from the epidermis that change their shape upon contact with the neuromasts.

2) Evidence showing that nBCare acting as a niche for the mantle cells to maintain their stemness is missing.

In the discussion part, the authors write "Our results suggest that during development, the neural lineage induces the formation of its own niche by fate conversion of neighboring epithelial cells." (subsection “Do nBCs constitute 1 a niche for neural stem cells?”). This statement is a bit surprising since no experiment in the result part are presented in this direction to prove that nBCs function as a "stem cell niche" for mantle cells. From my point of view, evidence is here totally lacking. Certainly the electron microscopy data "revealed that the membranes of border cells are intimately associated with those of mantle cells, often producing cytoplasmic protrusions into one another", yet this does not prove that these two cell types signal to each other and even less that the nBCs are necessary for the maintenance of the mantle cells as stem cells. The authors should address this point or remove the "stem cell niche" aspect from the manuscript, and also then change the title.

Reviewer #2:

Summary:

The manuscript contains many neat long-term experiments that lead to new insights into the generation and maintenance of neuromasts. The strong point in this paper is that organogenesis and cell turnover is followed over long time scales. However, some additional statistics, experiments and text changes are necessary to convincingly define these stem cell lineages.

Essential revisions:

– All Figures need meaningful statistics. In the current version of the manuscript the statistics sometimes refer to clones, sometimes neuromasts and it is not always clear how many fish were used and what age these fish were. This needs to be clarified over all Figures. It should be added to the Figures themselves and also be explained in more detail in the text. Furthermore, it should be explained what happens to clones that are not explicitly mentioned in the analysis. For example, for the lineage in Figure 3 it is stated that 45.1% of clones end up in the neural lineage while 12.5% end up in the neuromast border cell lineage.

However, it is not clear what happens to the rest. This should be explained here and in all other quantifications.

Especially Figure 3 and Figure 5 need additional quantification to assess all clones analyzed for these experiments.

– It would tremendously help in Figure 3, Figure 4 and Figure 6 to additionally show a DAPI staining to clarify where the exact boundaries of the respective organs are. This would help to appreciate the exact location of the labeled cells. In Figure 4 and Figure 6 DAPI panels would additionally help to differentiate between real cell ablation and mere photobleaching of the marker.

– As it is noted forFigure 6 that mantle cells first coalesce and then re-enter the cell cycle, some BrdU or EdU staining should be performed to support this claim.

– To underline the authors' findings in the live imaging approaches, an additional 4D reconstruction of the video should be shown. Z information would help to understand the source of the epithelial cells and when and how they interact with the neuromast cells.

– Along the same lines, the authors should add 4D imaging of the double K15 transgenic line that also labels the cell membrane. This would clarify how the presumptive neuromast border cells contact the neural cells and how and when the associated cellular changes occur. This would also then link this data to the EM data presented in Figure 2 in which the authors suggest that cells are in contact with each other and share desmosomes. Otherwise, this data is never mentioned again in the manuscript or the discussion and falls a bit out of focus of the study. If neuromast and epidermal cells interact and this interaction leads to cell shape changes, this would be a substantial addition to the findings the authors present. As such experiments in the adult fish would probably take a long time, this set of experiments could also be performed in the developing larvae. As the authors have access to confocal as well as light sheet microscopes these experiments should be feasible with the current molecular markers used in the manuscript and would massively help to clarify a potential role of protrusions and cell-cell contacts.

– It would be very interesting to know how the tissue reacts in case border cells are depleted (for example by laser ablation as done in other experiments). Will new border cells be recruited? Will other cells transform? This data will reveal the biological function of the newly defined neuromast border cells. These experiments could also be done in developing larvae for time reasons.

– Even though the authors did not explicitly find lineages that give rise to both, neural and border cells, this possibility cannot be completely ruled out. The text should be adjusted accordingly.

Reviewer #3:

Comprehensive analysis of lineage and fate with a variety of transgenic lines allows Seleit et al., to identify a *K15+* population as the previously elusive population of stem cells that serves as a source of source of both neural and epidermal lineages in the growing and regenerating Medaka posterior lateral line neuromasts. Their analysis suggests that while some *K15+* cells exclusively generate all the neural cells of the neuromast, others exclusively generate neuromast Border cells (nBC), a new epidermal sub-population defined by the authors that surrounds mantle cells and additional progeny that contribute exclusively to surrounding epidermal tissue. The authors also use EM to reveal intimate interactions between the mantle cells and the surrounding nBCs. The authors also emphasize how juxtaposition of a neural population, deposited by a migrating primordium, with its surrounding epidermal neighbors establishes interactions between cells of different embryonic origins to establish this stem cell niche. The results are clear and well laid out. The observations define a system that will serve well as a model for development of a stem cell niche in the future.

I found the EM results, which show potential interactions between mantle cells and surrounding nBC cells to be quite intriguing. To me they were reminiscent of observations made in the *Drosophila* testis where germline stem cells insert microtubule based nanotubes (cytoplasmic processes) into associated niche cells to receive Dpp signals, which are required for Germline stem cell survival. http://www.nature.com/nature/journal/v523/n7560/abs/nature14602.html In light of the *Drosophila* observations, I believe these observations in the neuromast are quite interesting, worth sharing, and should be explored further.

I did, however, have a difficult time initially understanding some genetic manipulations as I was not familiar with the Gaudi system. Some brief introduction to the Gaudi system would help. This is primarily a descriptive paper and will serve as an important baseline for future studies.

---

## [Author Response]

1) Reviewer 2 requests an additional video that includes a membrane label.

We followed the advice of the reviewer and performed additional imaging of skin epithelial cells during their conversion to nBCs. This data can be found in the new Figure 8—figure supplement 2, in the new Video 5, and in an additional video that we include in this response (Video-Reviewer, independent videos for membranes, nuclei and merge).

Our initial attempts involved the combination of transgenic lines and the imaging device suggested by reviewer 2 – Tg(*K15*:mYFP)(*K15*:H2B-RFP) imaged with the MuVi-Spim. This device is equipped with two 15x objectives, which provide a nice spatial resolution when combined with nuclear-tagged fluorescent proteins – data presented in Figure 8 and Video 2, Video 3 – but is not ideal when combined to membrane-tagged FPs. In the video movie attached to this response, we can observe that the morphological remodelling of K15 nuclei is accompanied by similar changes to membranes of induced cells, which we had already learnt from data presented in the previous version of the manuscript – Figure 8. However, the low spatial resolution of the device combined with the excess of membrane label both in the skin epithelium and in neuromast mantle cells of Tg(*K15*:mYFP) precluded us from addressing the possible existence and timing of protrusions from skin epithelial cells during their conversion to nBCs. We believe that this new dataset does not add to the revised version of the manuscript and we therefore do not include it.

Instead, we proceeded with a complementary strategy to tackle membrane dynamics during neuromast formation. To visualise membranes more adequately we followed a mosaic approach by injecting the *K15*:mYFP plasmid in 2-4 cell stage medaka embryos. This allowed sparse labeling of skin epithelial cells along the trunk (new Figure 8—figure supplement 3) that we imaged at three time points (day 8, day 9.5, day 13 post fertilisation) with a SPE Leica Confocal microscope equipped with a 40x immersion objective. This set-up provides a better spatial resolution, but is not suitable for long-term live-imaging as required to follow the induction event continuously. Since the K15 promoter drives expression both in skin epithelium and also in neuromast mantle cells, the approach allowed us to follow clones in both lineages. The data we present in the new Figure 8—figure supplement 3 shows that the skin epithelial cells do not extend obvious protrusions before, during or after their conversion to nBCs. We do however detect small protrusions similar to the ones observed in the EM once the organ is mature – new Video 5. However, when analysing*K15+* neural stem cell clones we consistently observed cytoplasmic extensions from the neural lineage. Iterative imaging of the same clones revealed that these processes are indeed dynamic and transient, since they disappear at early postembryonic stages – see new Figure 8—figure supplement 3’’. These new and unexpected data relates to the observation made by Reviewer 3, who suggested that the cytoplasmic protrusions between nBCs and *K15+* mantle cells observed by EM are reminiscent of the so-called nanotubes in the male germline of *Drosophila melanogaster*. We do not explore functional aspects of these protrusions in this study, but do speculate about these processes in the Discussion.

We thank the reviewers for this suggestion, which has helped improve our manuscript.

2) There was some discussion amongst the reviewers about experimental work that could be done to demonstrate a functional (niche) role for the border cells. As you will see, reviewer 1 comments on the current lack of experimental support for this interpretation, and ablation studies have been suggested by reviewer 2. It was acknowledged that these might be challenging to do, but it was felt that some attempt should be made to address this. It was also felt that unless further experimental support could be provided here, the authors should tone down their interpretations about the niche and alter the wording in the title.

We agree with the view from the reviewers, and have followed their suggestions to experimentally tackle a structural (niche) function for nBCs. Due to the lack of a permanent label for nBCs we have performed our experiments on Tg(*K15*:H2B-EGFP) and/or *neuromast^K8^*:H2B-EGFP during late embryogenesis, a stage in which most nBCs still retain the label expressed by their skin epithelial precursors. We have performed two sets of experiments to address the function of nBCs: a) 2-photon laser ablation of 1 or 2 nBCs per neuromast, b) 2-photon laser ablation of most nBCs of a neuromast. Both of them suggest that nBCs constitute a physical niche for *K15*+ neural stem cells in the neuromast, and we show these experiments in the new Figure 10 and in Video 6.

Briefly, when one or few nBCs are ablated in neuromasts we observe an immediate reaction from the neural lineage, where *K15*+ neural stem cells move internally as if they were detached from their physical anchor. This effect is certainly more drastic when most nBCs of a neuromast are ablated, and we show that this has massive structural consequences for the neural lineage. To explore whether the *K15*+ stem cells lose their stem cell marker or die, we iteratively imaged neuromasts after injury. We observed that the neural lineage is maintained and the integrity of the organ is re-established when new border cells re-appear, a process that happens within a week after injury. Our data suggest that these new nBCs originate from a re-recruitment event from the skin epithelium since some are positive for k15 promoter driven FPs, but we cannot exclude that they also originate from a highly proliferative nBC that was unlabeled and therefore not targeted during the 2-photon laser ablation. These new results from the requested experiments indicate that nBCs act as a physical niche for the K15 neural stem cells and that they are important for maintaining neuromast integrity and architecture. We have also expanded the Discussion in this regard.

An additional experiment suggested by reviewer 2 – that addresses a concern raised by reviewer 1 as well – is also relevant for this point. We explored other teleost fish and found that in all cases analysed (>10 neuromasts in *Poecilia formosa*, >10 neuromasts in *Danio rerio*) neuromasts contained nBCs that are adjacent to mantle cells – new Figure 9. We also generated a zebrafish Tg(ol*K15*:H2B-EGFP) and used it to demonstrate not only the existence of this cell type but also the conservation of the induction process. The new Video 6 shows that nBCs in zebrafish are induced from the skin epithelium upon the deposition of the neural lineage. Although not conclusive, the conservation of nBCs in distantly related teleost favours the view that nBCs are an integral part of the neuromast.

3) In the Discussion, there could be more emphasis on what our understanding from other model systems has revealed about what constitutes a stem cell niche, why it is important and how the neuromast might serve as a useful model for the exploration of their formation and maintenance. It might also help discuss what has been learnt so far in this system and what additional observations might need to be made to firmly establish the neuromast as a model for self-organization of a stem cell niche.

We have modified the Discussion to include the additional information suggested by the reviewers. In the revised version of the Discussion, we have included a more detailed description of what a niche is, and have provided examples of different niche complexity among the most studied models for post-embryonic stem cells. We have also highlighted what we believe is the main contribution of the neuromast to the field of niche-stem cell interaction and have stated our thoughts on future directions to explore this system further.

4) Please give all details of quantification and statistical analysis, as requested by reviewer 2.

We have followed the request of the reviewers and modified the way in which we present the quantifications of clones.

5) Please provide a brief explanation for the Gaudi methodology, as requested by two of the reviewers.

We have provided a better description of the Gaudi methodology as requested by the reviewers – first sentences on the “*nBCs constitute an independent life-long lineage*” section. We have also complemented the previous Figure 3 with a scheme of the genetic tools used to generate permanently labelled clones and how they are used in the neuromast model – Figure 3.

Reviewer #1:This is a very nice study, the lineage tracing data are very clear and the light and electron microscopy data very informative. In addition, the paper is very clearly written and pleasant to read. Yet, I have 2 major concerns about the manuscript.1) Evidence showing that nBCare an integral part of the neuromasts is missing.The authors describe the newly identified "nBC" as an integral part of the neuromasts. Since those cells have a different origin (the overlying epidermis, the authors show that very clearly), this is not clear to me at which point the nBC become part of the neuromasts. That the nBCare changing shape in the plane of the epidermis as neuromasts get deposited and mature is clear, but to my point of view, this is not sufficient to claim that they are an integral part of the neuromasts. If this is not the case, the nBCs should be considered as cells from the epidermis that change their shape upon contact with the neuromasts.

We appreciate that the reviewer finds this study informative and well written.

In the revised version of the manuscript, we have performed additional experiments and observations that in our view consolidate the notion that nBCs are indeed an integral part of neuromasts. Overall, we show:

a) Direct contact between nBCs and mantle stem cells – desmosomes in EM images, Figure 2.

b) A morphological and molecular change – the loss of K15 expression – upon recruitment.

c) Clonal expansion of nBCs surrounding the neural tissue that is stable life-long – both with ubiquitous and cell-specific drivers.

d) The presence of nBCs in every neuromast of medaka, regardless of their location in the body, their developmental origin and the time of their formation.

e) The presence of nBCs in neuromasts of other teleost fish, *Poecilia formosa* and *Danio rerio*, favouring the view that nBCs are present in every neuromast – new Figure 9.

f) Local structural responses of the neural lineage when 1 – 2 nBCs are ablated, and massive structural consequences when most of them are hit – new Figure 10 and new Video 7. Please also refer to the essential revisions, point 2.

g) The recovery of organ shape and distribution of cell types upon arrival of new nBCs – new Figure 10.

While this does not definitively prove nBCs are part of the organ we believe our data provide compelling evidence that they should be considered as such. More generally, we share with the reviewer the interest in what constitutes an organ. And indeed cases do exist where cells are considered part of an organ even when their morphology and expression profile is continuous with the surrounding tissues outside of it, and in the absence of physical contact – e.g. epithelial cells of the cornea are considered part of the eye. Besides the fact that nBCs are cells that display intimate contacts to neural lineage cells, they also have a major role in maintaining neuromast architecture – as shown in the new Figure 10. Since the neuromast is a mechanosensory organ, we believe that a cell type that directly impacts organ architecture is of critical importance for normal neuromast homeostasis and function.

2) Evidence showing that nBCare acting as a niche for the mantle cells to maintain their stemness is missing.

We have addressed this point experimentally and provide new data indicating that nBCs indeed constitute a physical niche for neural stem cells – new Figure 10. In the revised Discussion we have also included a list of the pre-requisites for a cell type to be considered a niche, and show that nBCsfulfil these criteria comparably with other well established niches. The reviewer can find a more detailed response in the points 2 & 3 of the essential revisions.

In the discussion part, the authors write "Our results suggest that during development, the neural lineage induces the formation of its own niche by fate conversion of neighboring epithelial cells." (subsection “Do nBCs constitute 1 a niche for neural stem cells?”). This statement is a bit surprising since no experiment in the result part are presented in this direction to prove that nBCs function as a "stem cell niche" for mantle cells. From my point of view, evidence is here totally lacking. Certainly the electron microscopy data "revealed that the membranes of border cells are intimately associated with those of mantle cells, often producing cytoplasmic protrusions into one another", yet this does not prove that these two cell types signal to each other and even less that the nBCs are necessary for the maintenance of the mantle cells as stem cells. The authors should address this point or remove the "stem cell niche" aspect from the manuscript, and also then change the title.Reviewer #2:Summary:The manuscript contains many neat long-term experiments that lead to new insights into the generation and maintenance of neuromasts. The strong point in this paper is that organogenesis and cell turnover is followed over long time scales. However, some additional statistics, experiments and text changes are necessary to convincingly define these stem cell lineages.Essential revisions:– All Figures need meaningful statistics. In the current version of the manuscript the statistics sometimes refer to clones, sometimes neuromasts and it is not always clear how many fish were used and what age these fish were. This needs to be clarified over all Figures. It should be added to the Figures themselves and also be explained in more detail in the text. Furthermore, it should be explained what happens to clones that are not explicitly mentioned in the analysis. For example, for the lineage in Figure 3 it is stated that 45.1% of clones end up in the neural lineage while 12.5% end up in the neuromast border cell lineage. However, it is not clear what happens to the rest. This should be explained here and in all other quantifications.Especially Figure 3 and Figure 5 need additional quantification to assess all clones analyzed for these experiments.

The reviewer is right in that the numbers were presented in a confusing manner, which we have improved in the revised version of our manuscript. Please refer to the response to the reviewer 1 to see an extended explanation of the clones involved in Figure 3.

We have followed the advice of the reviewer and modified the way in which we present the quantifications. The revised version shows a more consistent description of clones, and we have added the relevant numbers (numbers of clones/ neuromasts/ cluster/ fish) to the text and figure legends.

– It would tremendously help in Figure 3, Figure 4 and Figure 6 to additionally show a DAPI staining to clarify where the exact boundaries of the respective organs are. This would help to appreciate the exact location of the labeled cells. In Figure 4 and Figure 6 DAPI panels would additionally help to differentiate between real cell ablation and mere photobleaching of the marker.

We believe there is a confusion here. On fixed samples DAPI labels the nuclei of all cells, but in live embryos it only labels hair cells (the reviewer can check Seleit *et al.,* 2017). The experiments mentioned by the reviewer are all in vivo, and we do need the embryos to survive the imaging in order to analyse the short-term (Figure 3 and Figure 6) and long-term (Figure 4) effects of the treatment – be it cell labelling (Figure 3) or cell ablation (Figure 4 and Figure 6). Therefore, we cannot use DAPI to label all cells in the organ or organ boundaries.

In order to clarify the point that the reviewer raises, we have complemented the previous version of Figure 3 with a scheme depicting our way to identify cell types. The boundary of the neural lineage and nBCs is evident when using a bright-field view (new Figure 3—figure supplement 1). We show cases combining bright-field views with a transgenic line labelling just mantle cells and with another transgenic line labelling all nuclei. Nuclei with an elongated shape at the external boundary were annotated as border cells, nuclei internal of the boundary were annotated as mantle cells – data confirmed by 3D reconstructions and data presented in Figure 1 – and nuclei in the inner part of the organ were annotated as support cells or hair cells depending on their location and size.

As for whether the case in Figure 4 is a true ablation or mere photobleaching, we believe that the presence of both scar-tissue and puncta of the disassociated burst nuclei point towards a real cellular ablation. In addition, iterative imaging revealed that those cases resulted in a proper ablation, since the fluorescent signal was not recovered soon after the treatment – as it would be expected from mere photo-bleaching.

– As it is noted forFigure 6 that mantle cells first coalesce and then re-enter the cell cycle, some BrdU or EdU staining should be performed to support this claim.

Our observation of coalescence followed by mitotic activity was based on iterative imaging of the same organ during the regeneration period. We understand that our previous statement is strictly speaking incorrect – if a cell re-enters the cell cycle it is assumed that it had exited it, which is something we are not showing (in fact we have shown that mantle cells in medaka are cycling even under homeostatic conditions). We have therefore modified the sentence to more accurately reflect what we have learnt from the experiment.

– To underline the authors' findings in the live imaging approaches, an additional 4D reconstruction of the video should be shown. Z information would help to understand the source of the epithelial cells and when and how they interact with the neuromast cells.

The reviewer can find a detailed answer to this point in the essential revisions point 1, where we explain our different attempts to address membrane dynamics during the recruitment phase.

Regarding the changes in Z during the conversion from epithelial cells to nBCs, we have performed additional experiments to enhance the resolution in that axis, and present the data in the new Figure 8—figure supplement 2. We observed that recruited nBCs are continuous with suprabasal epithelial cells and stay in that same Z plane. We also present a 3D reconstruction of one clone – Video 5 – to show this continuity in a clearer way.

– Along the same lines, the authors should add 4D imaging of the double K15 transgenic line that also labels the cell membrane. This would clarify how the presumptive neuromast border cells contact the neural cells and how and when the associated cellular changes occur. This would also then link this data to the EM data presented in Figure 2 in which the authors suggest that cells are in contact with each other and share desmosomes. Otherwise, this data is never mentioned again in the manuscript or the discussion and falls a bit out of focus of the study. If neuromast and epidermal cells interact and this interaction leads to cell shape changes, this would be a substantial addition to the findings the authors present. As such experiments in the adult fish would probably take a long time, this set of experiments could also be performed in the developing larvae. As the authors have access to confocal as well as light sheet microscopes these experiments should be feasible with the current molecular markers used in the manuscript and would massively help to clarify a potential role of protrusions and cell-cell contacts.

Please find a detail answer above, essential revisions, point 1.

The EM data shows that cells of mature organs are in contact with each other. We do not know – and we do not claim – that the interactions observed in the EM images are the ones triggering the conversion of skin epithelial cells into nBCs. In the revised Discussion we now speculate that this interaction is a strong argument for a stem cell–niche interaction.

During the course of some experiments suggested by this and other reviewers, however, we closely followed the activity of membranes of neural stem cell precursors. We did observe (as the reviewer predicted) a dynamic and transient activity of membrane protrusions, but these were much longer than the ones observed by EM, and were not found in mature organs. We share the excitement of the reviewer regarding these newly identified membrane protrusions and this is indeed a line of research we will follow. We include a brief comment about them in the revised Discussion.

– It would be very interesting to know how the tissue reacts in case border cells are depleted (for example by laser ablation as done in other experiments). Will new border cells be recruited? Will other cells transform? This data will reveal the biological function of the newly defined neuromast border cells. These experiments could also be done in developing larvae for time reasons.

We have done the experiments suggested by the reviewer and present the new data in the new Figure 10. The reviewer can find a more detailed answer in the answer to the essential revisions, point 2.

– Even though the authors did not explicitly find lineages that give rise to both, neural and border cells, this possibility cannot be completely ruled out. The text should be adjusted accordingly.

We have adjusted the text accordingly stating that the existence of bi-potent cells cannot be excluded.

Reviewer #3:Comprehensive analysis of lineage and fate with a variety of transgenic lines allows Seleit et al., to identify a K15+ population as the previously elusive population of stem cells that serves as a source of source of both neural and epidermal lineages in the growing and regenerating Medaka posterior lateral line neuromasts. Their analysis suggests that while some K15+ cells exclusively generate all the neural cells of the neuromast, others exclusively generate neuromast Border cells (nBC), a new epidermal sub-population defined by the authors that surrounds mantle cells and additional progeny that contribute exclusively to surrounding epidermal tissue. The authors also use EM to reveal intimate interactions between the mantle cells and the surrounding nBCs. The authors also emphasize how juxtaposition of a neural population, deposited by a migrating primordium, with its surrounding epidermal neighbors establishes interactions between cells of different embryonic origins to establish this stem cell niche. The results are clear and well laid out. The observations define a system that will serve well as a model for development of a stem cell niche in the future.

We appreciate the positive comments of the reviewer.

I found the EM results, which show potential interactions between mantle cells and surrounding nBC cells to be quite intriguing. To me they were reminiscent of observations made in the Drosophila testis where germline stem cells insert microtubule based nanotubes (cytoplasmic processes) into associated niche cells to receive Dpp signals, which are required for Germline stem cell survival. http://www.nature.com/nature/journal/v523/n7560/abs/nature14602.html In light of the Drosophila observations, I believe these observations in the neuromast are quite interesting, worth sharing, and should be explored further.

This is quite an interesting point and in the revised version of the discussion we do include a reference to the nanotubes. We also performed additional experiments to address the presence of protrusions detectable under confocal microscopy. Although we did not find major cytoplasmic extensions in mature organs, we were surprised to detect numerous dynamic protrusions during organogenesis. These were clearly different in morphology from the reported nanotubes, and we briefly speculate in the revised Discussion about their putative role.

I did, however, have a difficult time initially understanding some genetic manipulations as I was not familiar with the Gaudi system. Some brief introduction to the Gaudi system would help. This is primarily a descriptive paper and will serve as an important baseline for future studies.

We have provided a better description of the Gaudi methodology as requested by the reviewers – first sentences on the “*nBCs constitute an independent life-long lineage*” section. We have also complemented the previous Figure 3 with a scheme of the genetic tools used to generate permanently labelled clones and how they are used in the neuromast model – Figure 3.